# Analysis of Convergent Validity of Performance-Based Activities of Daily Living Assessed by PA-IADL Test in Relation to Traditional (Standard) Cognitive Assessment to Identify Older Adults with Mild Cognitive Impairment

**DOI:** 10.3390/bs13120975

**Published:** 2023-11-27

**Authors:** María Paula Fernández, José Antonio Labra, Julio Menor, Eva Alegre

**Affiliations:** 1Department of Psychology, Oviedo University, Plaza de Feijoo, 33003 Oviedo, Spain; paula@uniovi.es (M.P.F.); jmenor@uniovi.es (J.M.); 2Department of Well-Being and Health, Town Hall of Villaquilambre, 24193 Villaquilambre, Spain; evaalegre@villaquilambre.es

**Keywords:** mild cognitive impairment, instrumental activities of daily living, convergent validity, everyday cognition, performance-based test, cognitive tests, older adults, debut in MCI, flowering of MCI

## Abstract

Difficulty in performing instrumental activities of daily living (IADLs) is currently considered an important indicator of cognitive impairment in the elderly. A non-experimental case–control investigation was conducted to assess the convergent validity of the PA-IADL with traditional (standard) cognitive assessment tests in its ability to identify adults with mild cognitive impairment. The analysis of the data was carried out by means of various multivariate statistical tests, and the sequence in its execution led to the conclusion that 8 of the 12 Tasks that make up the PA-IADL allow for the identification of people with mild cognitive impairment (MCI) to the same extent as traditional cognitive assessment tests and regardless of age. Age was found to be a moderating variable in the performance of the eight tasks; however, the results allow us to hypothesize that people with MCI experience a significant decline when it happens but thereafter, the deterioration that occurs does so at the same rate as the deterioration experienced by healthy people. They also allow us to hypothesize that the difference in the cognitive skills required by the eight functional tasks, and therefore also in the cognitive skills required by the traditional (standard) tests of a person with MCI compared to a person of the same age without MCI (Healthy), is approximately 10 years. These hypotheses have remarkable relevance and should be tested via longitudinal research. In the meantime, the results highlight the importance of the IADL assessment for the diagnosis of MCI as a complement to the standard cognitive assessment.

## 1. Introduction

In recent years, daily functioning as a diagnostic criterion for mild cognitive impairment (MCI) has been the subject of numerous studies [1,2,3,4,5]. A recent review highlighted that instrumental activities of daily living (IADLs) performance is sensitive to early cognitive decline but that many tasks that assess IADLs are so simple that they are unable to detect MCI, so it is important to have standardized tools in which tasks are novel, related to the real world, and can be applied in a clinical setting [6].

There is evidence that older adults with MCI are significantly more impaired in IADLs than healthy older adults [2,7,8,9,10]. In the early work of Petersen [11] and in the diagnostic criteria proposed by the International Working Group on MCI [12], it was stated that the activities of daily living should remain preserved or minimally affected. Subsequently, the United States National Institute on Aging and the Alzheimer’s Association presented a series of recommendations for the diagnosis of MCI, including that functional abilities must be preserved, although they recognize that people with MCI may make more mistakes or need more time to perform complex tasks of daily living than people without impairment [13]. Finally, in the Diagnostic and Statistical Manual of Mental Disorders (DSM) of the American Psychiatric Association (APA), in its fifth edition, the term MCI has been included as a minor neurocognitive disorder and includes, among its diagnostic criteria, that the individual must make more effort, use compensatory strategies, or make an accommodation to maintain independence in IADLs [14,15]. In any case, there is a consensus in considering the results of cognitive tests as diagnostic criteria for people with MCI [16,17], whereby psychometric data play an important role in comparing the results of cognitive tests with the normative data of the reference population [18].

An additional issue regarding the diagnosis of MCI is that it is a relatively unstable diagnostic entity [19,20]. Longitudinal studies show that not all people with MCI develop dementia and that a significant proportion revert to normal. For example, de Rotrou et al. [21] found that one year after an initial evaluation, 48% of the people with cognitive deficits in the first evaluation normalized their scores, that is, they performed the tests in the same way as the participants classified as normal in the first evaluation [22,23]. Gauthier et al. [24] point out that there are many factors that affect cognitive performance in older people in addition to neurodegenerative disorders, such as education, vascular risk factors, hormonal changes, psychiatric disorders, or the use of anticholinergic drugs, which may explain why some cases of MCI are reversible [25,26]. 

As noted above, the most recent research has highlighted the importance of recognizing and characterizing functional deficits in cognitive decline. Thus, it has been shown that MCI participants perform worse than healthy older adults on tests that assess performance-based functional capacity [3,7,10,27]. Longitudinal studies examining functional trajectories have shown that cognitively normal older adults who develop MCI have more everyday functional limitations at initial assessment (baseline) and more rapid functional decline than cognitively stable older adults without impairment [28]. It has also been shown that MCI participants with poor performance in IADLs (e.g., money management, meal preparation) are more likely to develop dementia than MCI participants without functional deficits [5,9,29]. In the same way, Reppermund et al. [30] found that informant-based functional capacity in cognitively normal individuals at baseline, particularly in activities with high cognitive demand, predicted MCI and dementia at follow-up (24 months). Overall, these results indicate that functional status is compromised in MCI and that it may be as important in predicting future cognitive decline as the diagnosis of MCI using traditional cognitive tests. In this sense, Burton et al. [31] found that performance in IADLs could predict current MCI status, and some authors have proposed that the ability to perform everyday tasks without difficulty might be a better predictor of future cognitive decline than traditional cognitive tests or biomarkers [32]. One of the reasons could be that traditional cognitive tests do not consider the self-initiation or self-regulation processes that take place in everyday life [33] or the multitasking abilities of everyday activities; rather, they assess a specific cognitive process in a structured and controlled laboratory context [6].

One of the most used procedures to evaluate the ability to perform IADLs are self-reports and proxy measures (e.g., a relative, friend, or caregiver). However, these procedures do not take into account the multidimensional nature of the tasks and do not reveal subtle differences in performance [34,35]. On the contrary, performance-based measures allow us to analyze the way in which the participant performs the task by segmenting its execution into steps [36].

Studies using performance-based tests to assess IADLs have found that people with MCI show poorer performance on specific tasks related to financial management [37,38,39], purchases [40,41,42], transportation use [27], medication monitoring [43], meal preparation [43], and prospective memory [27]. The worst performance in these tasks has been associated with the deterioration of certain cognitive processes such as working memory, cognitive flexibility and planning skills, episodic memory, and processing speed [27,38,42,44]. Menor and Labra [45] and Labra and Menor [44] developed a performance-based test (PA-IADL) composed of 12 tasks that allows for the evaluation of IADLs related to health care, administrative management, financial management, the use of transport, and meal preparation. The construct validity of this test showed that working memory predicted performance in tasks related to filling pill boxes, the control of medication, the payment of a workshop via a bank, money management, bus route planning, and preparing and cooking a recipe. Executive functioning (cognitive flexibility and planning skills) was associated with the tasks of filling pill boxes, the control of medication, bus route planning, and following a cooking recipe, while episodic memory was associated with recall, the recognition of ingredients in a cooking recipe, and remembering future actions [44]. Finally, crystallized intelligence was associated with two tasks (document management and the management of bank documents). 

Assuming that both the cognitive component and functional performance are necessary and complementary for the daily functioning and quality of life of older people and for the diagnosis of MCI, we propose carrying out a cross-sectional study of a case (MCI) and a control (Healthy) with a causal–comparative and predictive purpose with two main objectives:To study the convergent validity of performance in the functional tasks of the PA-IADL with performance in cognitive tests (the standard method of MCI diagnosis) in their capacity as tests for detecting MCI.To analyze divergences in the classification of the two detection methods and identify their vulnerability limits, as well as determine the power and limitations of both procedures.

These two objectives are made up of subordinate objectives that will be explained in the data analysis section.

In relation to the discrimination capacity of the tasks that make up the PA-IADL, we hypothesized that the tasks that best discriminate between Healthy and MCI older adults are those that combine working memory and executive processes as well as episodic memory. However, no significant differences are expected between Healthy and MCI groups in tasks related to crystallized intelligence since these tasks reflect the knowledge that elderly individuals have due to their education and cultural level. This type of knowledge is expected to be conserved in the MCI group.

## 2. Materials and Methods

### 2.1. Sample Selection

All participants were recruited in social centers for the elderly in Asturias (Mancomunidad Comarca de la Sidra and Oviedo) and León (Villaquilambre). The managers of the centers were contacted to inform them about the purpose of the study and to request collaboration in its implementation. Center managers that agreed to collaborate provided the research team with the informed consent of 265 individuals over 60 years of age who had voluntarily decided to participate and who met the inclusion criteria: living independently at home, not needing or receiving any dependency care resources or family support to carry out their activities of daily living, and attending weekly social centers on their own to participate in workshops for active and healthy aging. Meeting at least one of the following criteria was grounds for exclusion: psychiatric diagnosis, neurological diagnosis, moderate or severe cognitive impairment (MMSE score less than 24), traumatic brain injury, central nervous system infections, medication for depressive or psychotic problems, visual deficit, and significant language comprehension deficit. Thus, 13 individuals were eliminated, and the remaining 252 individuals underwent a thorough evaluation (see Figure 1).

### 2.2. Measures

#### 2.2.1. Sociodemographic Data 

Data related to age, gender, marital status, and years of study were collected. The variables of age, gender, and years of study were used for case–control matching. Age was also considered in the different statistical analyses as a classification variable. 

#### 2.2.2. Screening Tests

Two screening tests were used: the Mini-Mental State Examination (MMSE) [46], whose scores were corrected for age and educational level according to the Spanish adaptation (alpha coefficient = 0.94; test–retest reliability = 0.87) [47] and the Geriatric Depression Scale (GDS) (alpha coefficient = 0.94; test–retest reliability = 0.85) [48]. A score below 24 on the MMSE indicates suspected dementia, and a score greater than 10 on the GDE is an indicator of depression.

#### 2.2.3. Cognitive Assessment

The test battery used to assess cognitive functions was selected based on the availability of normative data in the Spanish population. To assess episodic memory, both immediate and delayed, the Word List I and II tests of the Weschler Memory Scale were applied (alpha coefficient = 0.70–0.90) [23,49,50,51]. For processing speed, we used the Symbols and Digits Test (SDMT) (test–retest reliability = 0.93) [52,53]. Working memory was assessed using the Letters and Numbers tests of the Weschler Memory scale, as well as Arithmetic (two-halves reliability = 0.82; test–retest reliability = 0.86) and Digits (two-halves reliability = 0.87; test–retest reliability = 0.83) of the Wechsler Adult Intelligence Scale (WAIS-III) [54]. 

#### 2.2.4. Performance-Based Test to Assess Instrumental Activities of Daily Living (PA-IADL)

This performance-based test [45] consists of twelve tasks grouped into six areas (the use of medication and health arrangements, administrative arrangements, financial arrangements, meal preparation, the use of transportation, and shopping) that assess different complex daily activities using real-life materials (medicine bottles, pills, bank book, etc.). T1: Fill pill boxes following the instructions of a medical report. T2: The delayed recall of two observations from the medical report read at the beginning of the evaluation. T3: Control of medication (know for how many days there are pills left in the medicine bottle). T4: An event-based prospective memory task (request an appointment sheet when the evaluator indicates that a task is completed). T5: The payment of a workshop via a bank account. T6: Documentation management (choose one application form among several and locate the requested information). T7: Time-based prospective memory task (perform the action indicated after 15 min). T8: Money management (provide change after shopping at a supermarket). T9: The management of bank documents (know how much money is in the bank before and after paying an electricity consumption bill). T10: Cooking recipe preparation. T11: Bus route planning. T12: Recall/recognize the ingredients in task 10.

The evaluator should observe how the tasks are solved and record the corresponding score for each step on the recording sheet. All tasks are part of a story with two main characters so that aspects such as delayed memory or prospective memory can be assessed. The instrument was constructed on the basis of functional capacity to perform frequently performed tasks of daily life and not on the basis of performance at pencil-and-paper tasks oriented to the measurement of specific cognitive abilities. The instrument was validated in a sample of healthy elderly people without cognitive impairment, obtaining an internal consistency with a Cronbach’s α of 0.82. The construct validity analysis showed that the different tasks are grouped into three main factors: tasks related to fluid cognitive processes (working memory, reasoning, and planning), tasks related to crystallized cognitive processes, and tasks related to episodic memory processes. The inter-rater reliability was 0.99 for the fluid and episodic factor, 0.95 for the crystallized factor, and 0.99 for the total score. The test–retest reliability was 0.64 for the fluid and episodic factor, 0.63 for the crystallized factor, and 0.75 for the total score [44].

#### 2.2.5. Procedure 

Four psychology experts, blind to the research objectives, were adequately trained in the application of all the evaluation instruments. The results were pooled and differences, if any, were discussed until a consensual solution was reached. Four evaluation sessions were held for each participant. In the first, sociodemographic data were collected and screening tests performed. In the second and third sessions, the cognitive assessment was performed, and in the fourth session, the 12 tasks comprising the PA-IADL were administered. The time with each participant in each session varied between 45 and 60 min. All the evaluations were carried out in social centers for the elderly between 10:00 and 13:00 in the morning. On 32 occasions, it was necessary to alter the day and time of the evaluation session for some participants due to unforeseen (and therefore random) reasons [55]. For this reason, no data were lost.

#### 2.2.6. Definition and Selection of Cases and Controls

Once the assessment was carried out, the Cases were selected. Individuals who showed altered cognitive test scores (episodic memory, processing speed, and working memory), i.e., who deviated below the normative 1.5 SD group, were tentatively considered as Cases or MCI [12]. The final decision also considered the recommendations of the United States National Institute on Aging and the Alzheimer’s Association [13]. The aim was to select cases without moderate or severe cognitive impairment or other neurodegenerative disease that were clearly distinct from what would constitute a healthy person, who would be suitable to be a Control participant. Thus, 32 people with MCI were selected, representing 12.7% of the total sample evaluated. Similar prevalence rates have been found in other studies [56]. Once the Cases were selected, for each Case, a Control was sought who had to have the same age, sex, and years of study. Finally, the total sample consisted of 64 people aged between 60 and 98 years (mean age = 76.80; SD = 9.31); 79.68% (n = 51) were women and 20.31% (n = 13) were men (see Appendix A). Hereafter, we will refer to the variable experimental group (EG) to refer to the two subgroups, Cases (or MCI) and Controls (or Healthy individuals).

#### 2.2.7. Evaluation of the Effectiveness of the Pairing

No statistically significant differences were found between Cases and Controls in the matching variables, age, sex, and years of study or on the Geriatric Depression Scale (see Appendix A to observe this aspect in detail).

However, there was remarkably different behavior between the two experimental groups in the indicator variables of cognitive performance. Table 1 shows these differences numerically and graphically (see also the descriptive statistics in Appendix A). It can be seen that the two experimental groups showed very different behavior in the global cognitive functioning variable evaluated through the MMSE. Although the differences between the means were statistically significant (F_BF_ = 6.11; *df*_1_ = 1, *df*_2_ = 48.9; *p* = 0.017; η^2^ = 0.090), the mean and the median values presented minimal differences and were always within the range of normal cognitive functioning. What was significant, from our point of view, was not the difference between the means but the difference in the distribution of the variable. In Healthy participants, there was a negative bias (λ_1_= −1.72), that is, most healthy people had excellent global cognitive functioning, and only some people had mildly poor functioning due to age (ages above 80 years), but in the sample with MCI, the distribution was more homogeneous (despite also being distributed non-normally, both skewness and kurtosis statistics were less than 1). Another aspect to highlight is that the measure of variability (SD and CV) was greater in the MCI group, that is, there was great variability among the MCI participants in terms of cognitive functioning evaluated via the MMSE, possibly due to the differences between the various affected domains of these individuals [12,57].

Regarding the 6 cognitive assessment measures (CgAs) CgT1-CgT6, statistically significant differences were observed in all the variables except CgT1 (as others have found or as is expected), and differences, although marginal, were observed in CgT5. However, and in the same way as it happens with the MMSE, the difference in the distribution of these two variables and the other four is very notable. This can be seen in the graphs inserted in Figure 2 and from observing the descriptive statistics of the distribution in Appendix A.

Thus, it can be concluded that the two experimental groups, Cases and Controls, did not differ in the sociodemographic variables or the indicator variable of emotional health; however, they had significantly different behavior in the indicator variables of cognitive health, MMSE, and the CgAs (CgT2, CgT3, CgT4, CgT6, and CgT7). Therefore, the composition of the two experimental groups was adequate for carrying out a cross-sectional case–control study with the aim pursued.

Finally, the total sample consisted of 64 individuals aged between 60 and 98 years (M = 76.80; SD = 9.31), of which 79.68% (n = 51) were female and 20.31% (n = 13) were male.

### 2.3. Data Analysis

Each of the two previously written objectives (see the Introduction) implies subordinate objectives that must be carried out in a certain order, and the analysis of the data is conditioned by the activities involved in them. The detailed objectives are as follows. First, the two objectives are described in detail, and then the data analysis sequence involved in carrying out each objective is defined.

**Objective** **1.**
*The study of the convergent validity of PA-IADL functional tasks with cognitive assessment tests (the traditional MCI detection method), hereinafter CgAs, in their capacity as MCI detection tests. This contains two subordinate objectives.*


Subordinate objective 1.1: To determine which PA-IADL tasks are useful for identifying and differentiating participants who have been classified as MCI and participants who have been classified as Healthy via CgA tests. Because all PA-IADL tasks require cognitive strategies for their performance and cognitive skills are naturally affected by age, to achieve this subordinate objective, it was considered necessary to determine two aspects. First, we assessed which tasks are useful for discriminating between MCI and Healthy individuals regardless of the age of the participants. Second, we assessed whether there were differences between MCI and Healthy participants in their performance according to age. The result of these analyses will allow us to understand which tasks have the greatest power to discriminate between MCI participants and Healthy ones, and we will call these target tasks (TTs).

Subordinate objective 1.2: To evaluate the effectiveness of the TT set in classifying the participants of the sample set into clearly differentiated categories.

**Objective** **2.**
*To analyze the divergences in the MCI-Healthy classification of the two methods, using the functional tasks of the PA-IADL (only the TTs) and through the CgAs in the diagnosis of MCI. This contains two subordinate objectives.*


Subordinate objective 2.1: To examine the differences in the performance of the TTs of the PA-IADL between the participants that make up the four groups of the confusion matrix, with particular interest in the participants that the two methods classify in different groups.

Subordinate objective 2.2: To examine what characteristics are shown by the participants that the two methods classify in different groups in the variables used to carry out the matching.

The data analysis carried out to achieve each of the two subordinate objectives contained in Objective 1 and Objective 2 was as follows.

#### 2.3.1. Data Analysis in Pursuit of Subordinate Objective 1.1

At this point, it is a priority to understand whether it is possible to study the 12 tasks together or, on the contrary, whether this must be done individually. The statistical advantage of multivariate tests over univariate tests is that they keep Type I error under control [58,59], but there is also a substantive advantage that in this particular case is very relevant, and that is that they would allow for a holistic study to be carried out using the integrated PA-IADL package in the same way that cognitive tests in the diagnosis of MCI constitute a holistic package. To achieve this, we began by evaluating the correlation in the performance of the 12 tasks with each other (see Appendix A), and it was found that the 12 tasks are intercorrelated. Specifically, our findings were as follows: 89.39% (n = 59) of the correlations were statistically significant; 45% (n = 30) of the correlations had a magnitude greater than 0.40, the minimum value for applying a multivariate methodology [60]; no correlation reached the value of 0.80, and therefore, it is unlikely that there is multicollinearity [61,62]; and Bartlett’s sphericity test allowed us to reject the null hypothesis that the residual covariance matrix is proportional to an identity matrix (χ^2^= 371.42; df = 77; *p* = 0.001). For all these reasons, it was concluded that it is more appropriate to use a multivariate methodology than a univariate one [58,63].

Thus, it was decided to first carry out a MANCOVA, taking age as a covariate, and then a MANOVA (EG × Age groups), introducing age as a classification variable. The two analyses provided us with complementary information. We will outline this below briefly.

Given that age has a statistically significant linear relationship with all tasks (except task 12) (see Appendix A), it is a priority to determine the tasks in which the difference in performance between Cases and Controls disappears when controlling for the effect of age and the tasks in which the differences between the two groups exist even when controlling for the effect of age. This is the reason that justifies the MANCOVA, and we began by checking whether the rest of the assumptions it requires were satisfied. It was verified that there was no relationship between the covariate and the assigned independent variable (EG), and it was also confirmed that Tasks 3 and 7 show non-parallel slopes in the groups of Cases and Controls. Based on this result, Task 12 was examined using an ANOVA; for the study of Tasks 3 and 7, the regions of non-significance and significance were determined using the Johnson–Neyman technique [64] and by performing the ANCOVA in the regions of significance; and a joint analysis of the remaining 10 tasks was performed using the MANCOVA, following the modeling strategy proposed by Milliken and Johnson [65]. Since the difference in the execution of Tasks T2, T6, T8, and T9 between the EGs disappears once the effect of age is controlled, these four tasks did not form part of the subsequent MANOVA.

Because the sample of participants covered ages between 60 and 98 years, for practical purposes, in addition to understanding whether age influences the performance of the different tasks, it is of great interest to understand whether performance in the groups of Cases and Controls is moderated by age. To determine this, the sample was segmented into three age ranges (AR) (60–69, 70–79, and over 80 years), and a MANOVA (2 × 3) (EG × AR) was performed on the set of 8 tasks in which, once the effect of age was controlled, the differences between the EGs persisted. The result was examined using Wilks’ lambda statistic because it is the most robust test when the assumption of the homogeneity of the covariance matrices is not satisfied [66], as in this case (Box’s M =143.53; *g1* = 78 and 12,138.7; *p* = 0.005; Levene’s test detected non-homogeneous variances in tasks 4, 5, 11, and 12). Since the interaction was statistically significant, we proceeded to study the simple effects and their corresponding mean differences (Bonferroni correction). The analysis was completed by performing a stepwise discriminant analysis to examine which combination of dependent variables had the strongest power to differentiate between the 6 groups resulting from the combination (EG × AR), paying special attention to the standardized coefficients and the magnitude and sign of the centroids. The analysis was first performed on a randomly chosen training sample (70% of cases, approx. n = 50) and then on the whole sample (it was decided to not perform the analysis on the validation sample because 30% of the sample, the remaining sample, is too small to be useful as a validation sample).

The reference values are the *p*-value (α = 0.05) and the partial eta squared (η^2^) (0.01 small, 0.06 medium, and 0.14 large). See [67,68] and (1−β) > 0.80.

To aid in the interpretation of the results obtained via both the MANCOVA and MANOVA, the linear relationship between task performance and age was plotted, and the line of fit was also calculated using iterative weighted least squares (Loess, using the Epanechnikov kernel function in which data located near a particular point receive higher weights than data that are further away).

#### 2.3.2. Data Analysis in the Achievement of the Subordinate Objective 1.2

The previous analyses led to the conclusion that there were 8 TTs. Because the classification techniques are exploratory techniques and due to the small sample size, it was considered necessary to control the potential overfitting that could occur and also to control the effect of the method used for classification. Thus, the analysis was carried out using 5 different methods: the hierarchical cluster procedure using two methods, the furthest neighbor method and the Wald method—(both were chosen because they are robust to outlier scores, a probable case in such a small sample); the two-stage cluster method, the k-means method; and the discriminant method. The last two require indicating the number of clusters and the hierarchical procedures do not, and it is optional in the two-stage cluster analysis (in this case, it was performed without determining the number of clusters). Furthermore, the analysis was also performed via cross-validation, randomly selecting a new training sample (approximately 70% of cases, n = 42). Again, due to the small size of the validation sample (n = 22), the second analysis was performed on the full sample.

The percentage of coincidence was calculated between the MCI-Healthy classifications (i.e., Cases–Controls) carried out with the 5 classification techniques performed based on the TTs and the original classification carried out using the CgA tests. The magnitude of the agreement was also calculated using the Kappa coefficient.

#### 2.3.3. Data Analysis in the Achievement of Subordinate Objective 2.1

The decision was made to take as a reference the classification resulting from the hierarchical analysis using the Ward method and the resulting classification using the k-means method as these were the classifications that had the highest and lowest concordance between the two methods. The differences between the 4 groups of the confusion matrix in the 8 TTs were tested using the robust Brown–Forsythe ANOVA due to the heterogeneity of the variances between the groups, caused in part by the different sizes of the groups of the confusion matrix [69].

#### 2.3.4. Data Analysis in the Achievement of Subordinate Objective 2.2

This was accomplished by looking at the descriptive statistics. The analysis of the two subordinate objectives allowed us to identify the power, limitations, and vulnerability limits of the detection procedure based on cognitive tests.

The most crucial factor in a case–control study is to choose “the cases” and “the controls” Carefully [70]. If the selection is not correct and if the matching is not accurate, there will be selection bias, and the inferences will not be valid. This is one of the reasons that justifies this section being so extensive and so detailed. Much effort was invested in correctly identifying cases and controls, with the sole objective that the Cases were indisputably cases and that the Controls were indisputably controls. For this reason, the sample size was only 64 people in total. On the other hand, the data analysis that was carried out was in-depth and very orderly. As can be seen, each objective was carried out with different analyses as a “sensitivity analysis” to qualify the results and safeguard their replicability.

All analyses were performed using the statistical package IBM SPSS 27.

## 3. Results

For speed of reading, the detail of the magnitude of the statistics is omitted in the text, referring the reader to the specific table in which they are presented. With the intention of including all the information found in the analysis and due to logical reasons of space, additional information is attached in the Appendix A.

**Objective** **1.**
*The study of the convergent validity of the PA-IADL functional tasks with the cognitive tasks (traditional MCI detection methods) in their capacity as MCI detection tests.*


**Subordinate objective** **1.1.***To determine which PA-IADL tasks are useful for identifying people who have been classified as MCI using cognitive tests.* The results are reported in Table 2 and Table 3 and Figure 3 in the text and in Appendix A.

The MANCOVA revealed that once the effect of age was controlled, there were statistically significant differences between Cases and Controls in the set of tasks (Λ = 0.577; F = 4.31; gl1 = 9, gl2 = 53; p = 0.001; η^2^ = 0.423; 1−β = 0.993), but only tasks T1, T4, T5, T10, and T11 were responsible for the result achieved. The ANOVA revealed that the two EGs performed significantly differently at T12 (see Appendix A).

The region of non-significance calculated using the Johnson–Neyman technique for the performance of tasks T3 and T7 occurs from 80 years (see Figure 3 in the text and Appendix A, which support the result of the calculation). The ANCOVA (see Appendix A) performed in the three age ranges shows that once the effect of age was controlled, there were statistically significant differences between Cases and Controls in the performance of task 3 in the age ranges 60–69 and 70 −79 years and in the performance of task 7 in the age range 70–79.

In all the statistically significant contrasts of the MANCOVA, ANOVA, and ANCOVA, the effect size was large.

The MANOVA (EG × AR; 2 × 3) revealed that the interaction was statistically significant (Λ = 0.595; F = 1.88; gl1 = 16, gl2 = 1042; p = 0.030; η^2^ = 0.229; 1−β = 0.934), and therefore, it can be concluded that the difference between Cases and Controls in the set of tasks T1, T3, T4, T5, T7, T10, T11, and T12 is moderated by age range. Table 2 shows that all the simple effects of each AR and each EG were statistically significant, and the magnitude of the effect was large.

On the other hand, in the MCI group, Cases, the study of the differences in means showed that the performance in tasks T3 and T12 remained constant in the three age segments. It also revealed that participants aged 60–69 differed from participants aged 70–79 and from participants over 80 years of age in performance at T1, T4, T10, and T11, but in these tasks, there were no statistically significant differences between 70–79 and >80 years. That is, the greatest deterioration occurred between 60–69 and 70–79 years, which is where the greatest change was. Later, from that age, the evolution of deterioration is slower. Differences were only observed between 70–79 and >80 years in the performance of T5. Task 7 only showed differences between the most distant age groups (60–69 and >80 years). Figure 3 in the text, and Appendix A, show that the deterioration of the trajectory in these tasks was significantly smoother in the control group.

In the Control group (Healthy) the performance in tasks T1, T5, T10, T11, and T12 was not significantly different in the three age segments, and tasks T3, T4, and T7 showed different or significantly deteriorated performance from 70–79 years. In other words, the deterioration that occurred in these three tasks in the Control participants in the first age group studied was very small. If we look back at Figure 3 (in the text) and Appendix A (in the Appendix A), this numerical result is even more revealing.

Although they are only for descriptive purposes, it is beneficial to look at the graphs shown in the Appendix A to understand in a more complete way how the differential performance in the tasks occurred in Healthy and MCI participants in the three age groups considered. It can be seen how age influenced the performance of all of these tasks, and it was always worse in the Case group. However, the rate at which deterioration occurred with age was very different in them. In tasks 12, 10, and 4 the difference in performance between Cases and Controls was of a more or less constant magnitude throughout the age range, but age had greater virulence in T4 than in T10, and it had much less virulence in T12. Performance in tasks T3 and T7 deteriorated considerably as age advanced, and at the age of over 80 years, the Cases and Controls did not differ in their performance; however, skill in the two tasks was considerably reduced in participants with MCI in younger ages. The opposite happened in tasks T5, T1, and T11. The performance in these tasks was similar between the ages of 60 and 69 years, slowly deteriorating with advances in age, although more markedly in the MCI group, distancing itself abruptly at the age of 80 years.

The previous results allow us to conclude that the target tasks are tasks T1, T3, T4, T5, T7, T10, T11, and T12.

The discriminant analysis (see Table 2) provides complementary provides complementary information to the result revealed by the MANOVA. A variable was created to identify the six groups resulting from the interaction between the levels of the variables (EG × AR; 2 × 3). The “stepwise” analysis allowed us to capture the relevance of two tasks as the tasks most responsible for the differentiation between the six groups. In the training sample, they were tasks T4 and T11, and in the total sample, they were T4 and T12, classifying 60% of the cases correctly in the first analysis in and 50% in the second. Table 2 shows all the details of the statistical results. The most relevant is the reading of the centroids in which both the analyses performed on the training sample and on the total sample converge on the result. The centroids in the Case participants (MCI) whose ages were in the ranges 70–79 and >80 years were negative and of similar magnitudes. The centroids of the Control participants (Healthy) whose age range was between 60–69 and 70–79 years were positive but far apart in magnitude. On the other hand, the centroid of the Control participants >80 years was also negative, but it was practically the same distance from the Control participants of 70–79 years as from the Cases >80 years. Finally, Case participants aged 60–69 years were positive and very close in magnitude to Controls aged 70–79 years. Table 3 shows this result graphically (see also Appendix A) which, again, helps us to appreciate at a glance the magnitude of the previously explained statistical result. The closeness can be appreciated, on one hand, between Case participants (MCI) of the age ranges 70–79 and >80 years and, on the other hand, between the Control participants of 60–69 and 70–79 years. It can also be seen that the youngest MCI participants (60–69 years old) were very close to Healthy people aged 70–79 years.

**Subordinate objective** **1.2.***Evaluate the effectiveness of the set of target tasks to classify the sample participants into clearly differentiated categories.* The results are presented in Table 4 and Table 5 in the text and in Appendix A.

The results of the five exploratory classification techniques were conclusive. Table 4 (see also Appendix A) shows that the percentage of coincidence in the classification of Cases and Controls between the cognitive tests and the functional tasks found in the total sample oscillated between 76.56% of the k-means procedure (Kappa = 0.531) and 81.24% (Kappa.625) of the hierarchical cluster procedure (Ward’s method). The other three procedures had the same percentage of coincidence (70.68%; Kappa = 0.594).

Appendix A shows numerically (Appendix A) and graphically (Appendix A) the joint strength of the eight8 tasks in classifying the participants in the other five exploratory techniques into differentiated groups. It can be seen that all the procedures differentiated between the two groups, none found ungrouped cases, and the number of misclassified cases was similar in all of them. Details of the results found are highlighted at the bottom of the figures.

**Objective** **2.**
*To analyze the divergences in the MCI-Healthy classification of the two methods, the functional tasks of the PA-IADL (only the target tasks) and cognitive tests, in the diagnosis of MCI.*


**Subordinate objective** **2.1.**
*To examine the differences in the performance of the PA-IADL target tasks among the participants that make up the four groups of the confusion matrix, with special interest in the participants that the CgAs and the eight tasks of the PA-IADL classified into different groups. The results are shown in Table 5.*


Table 5 shows great differences in the performance of all the tasks between the groups of participants that both procedures classify as Healthy (Healthy_C_-Healthy_P_) and the groups that both procedures classify as MCI (MCI_C_-MCI_P_). The differences were statistically significant. Again, this demonstrates, on one hand, the usefulness and power of the eight tasks to differentiate between people with and without MCI and, on the other hand, the existence of convergent validity between the two modes of detection.

Regarding the people that the two approaches classified differently, it can be seen from observing the means that the people classified by the CgA tests as Healthy who the PA-IADL TT classified as MCI (Healthy_C_-MCI_P_) had a similar performance to the individuals that both approaches classified as MCI (MCI_C_-MCI_P_), and that the people classified by the cognitive tests as MCI who the PA-IADL classified as Healthy (MCI_C_-Healthy_P_) had similar performance to the people who both approaches classified as Healthy (Healthy_C_-Healthy_P_). None of these contrasts were statistically significant.

On the other hand, it was shown that the difference in this analysis between the two cluster methods, which have greater and lesser agreement with the classification made by standard cognitive tests, was very subtle and practically irrelevant. They fully coincide in the Healthy_C_-Healthy_P_ and Healthy_C_-MCI_P_ classifications, and slightly diverge in the MCI_C_-MCI_P_ and MCI_C_-Healthy_P_ classifications, the only difference being that k-means categorized three more people as MCI than the hierarchical procedure (Ward’s method) (see Appendix A).

**Subordinate objective** **2.2.**
*To examine what characteristics the participants classified in non-convergent categories show by the two methods in the variables used to perform the matching. This is carried out by looking at descriptive statistics.*


For the analysis of the characteristics of the participants classified in non-convergent categories, those obtained via the method with the lowest percentage of coincidence of Cases and Controls (k-means clustering) were taken into account.

The six people who were classified as Healthy individuals by the CgA tests and as MCI by the PA-IADL (Healthy_C_-MCI_P_) were between 81 and 94 years old (M = 86; SD = 5.29); five were female and one was male, and they had completed a mean of 3.66 years of study (SD = 1.36). On the other hand, the nine people who were classified as MCI individuals by the CgA tests and as Healthy by the PA-IADL (MCI_C_-Healthy_P_) were between 60 and 83 years old (M = 67.6; SD = 9.20); six were women and three were men, and they had completed a mean of 7.44 years of study (SD = 1.33). Therefore, the two non-convergent categories differed fundamentally in terms of age and years of study. The participants in the first category were older and had fewer years of study than those in the second category. This aspect requires an in-depth study to make sense of these results.

## 4. Discussion and Conclusions

A case–control study was carried out with two fundamental objectives: firstly, to analyze to what extent the performance in the functional tasks of the IADL-PA and the cognitive tests converge in their ability to detect MCI in people without pathologies, without moderate or severe cognitive impairment, and who live independently in the community, and secondly, to analyze the divergences in the classification of the two detection methods and identify their limits of vulnerability, as well as determine the power and limitations of both procedures.

Both objectives were aimed at evaluating the convergent validity of performance-based activities of daily living assessed by the PA-IADL test in relation to traditional (standard) cognitive assessments to identify older adults with mild cognitive impairment, which gives a name to the title of this article. In this section, we discuss the results found in the order in which the data analysis was planned, and in this way, in addition to discussing the results found, we discover new hypotheses that are proposed in light of these and the conclusions generated. Thus, we organize the most significant results and their corresponding analysis into four points.

### 4.1. The Influence of Age on Task Performance (IADL-PA) and the Differences between Cases and Controls after Controlling for the Effect of Age

Age was found to influence the performance of all tasks in the Cases and Controls. However, the influence of age is not the same in all of them. Its effect can be summarized in three points.

First, Cases and Controls differ statistically significantly in task 12. However, age does not explain this difference at all.

Second, age is the only variable that explains the difference between Cases and Controls in the execution of tasks T2, T6, T8, and T9. For this reason, once the effect of age is controlled, the differences between Cases and Controls in these four tasks disappear.

Third, age is a variable that explains in some the difference between Cases and Controls in executing tasks T1, T3, T4, T5, T7, T10, and T11. In fact, once the effect of age is controlled, the differences between Cases and Controls persist. Therefore, other variables that make the Cases and Controls different explain these differences.

Three conclusions emerge from these results.

First, age is not a decisive or significantly relevant predictor in classifying healthy individuals and those with mild cognitive impairment. Abundant studies support the notion that MCI can manifest as a comorbidity associated with other pathologies, diabetes, cancer, etc., and not necessarily as a product of age [71,72,73,74]. 

Second, although age explains the differences between the Cases and Controls in tasks T2, T6, T8, and T9, it must not be ruled out that the small sample size is responsible for the lack of test power to detect differences between both groups once age was controlled. We think this because if we look at Appendix A, we see that in these four tasks, both groups have apparently different behavior in the age range between approximately 70 and 79 years and a behavior similar to older age and at a younger age (except perhaps for T2). A possible explanation for these results could be that tasks T6 and T9 are tasks in which administrative and banking documentation must be handled and have a crystallized component, that is, they require domain-specific knowledge acquired through practice throughout life. They are, therefore, tasks in which people with mild cognitive impairment can make up for their deficits with acquired knowledge resulting from their daily practice through compensatory strategies [75]. 

The use of compensatory strategies in everyday life has been the subject of studies [76,77,78], and results suggest that their use seems to improve performance in everyday tasks in older people with poorer cognitive performance [79]. Joly-Burra et al. [80]. analyzed the selective optimization by compensation (SOC) model in a sample of people over 65 years of age and concluded that the use of compensatory strategies was more directly related to the cognitive performance of the participants and less so to age. It would be of interest in the future to delve into how the use of compensatory strategies affects daily tasks in people with MCI.

Third, if age does not explain the differences in the execution of task 12 by Cases and Controls, and if age is not the only variable that explains the differences in execution between the Cases and Controls in tasks T1, T3, T4, T5, T7, T10, and T11, the following hypotheses arise: Is age a moderating variable? That is, are the differences in performance between Cases and Controls in these tasks moderated by age? If we segment age into 10-year intervals (for example), would the differences between Cases and Controls be of the same magnitude in the same age ranges in all tasks? Is there a critical age range in the debut of MCI?

### 4.2. The Differential Behavior between Healthy and MCI Individuals Depending on the Age Range

The results found with the MANOVA allow us to conclude that performance in tasks T1, T3, T4, T5, T7, T10, T11, and T12 is different in Cases and Controls in the three arbitrarily formed age ranges. From these results, five conclusions emerge that must be discussed, and new hypotheses are proposed that open new lines of research. We consider the following.

First, age has a moderating effect on the performance of tasks T1, T3, T4, T5, T7, T10, T11, and T12, and therefore, they are performed with different degrees of solvency in healthy people and in people with MCI in the three ARs. As highlighted in the introduction, this aspect has been widely documented.

Second, at 60–69 years (AR1) of age, people with MCI show performance similar to that of healthy people aged 70–79 years (AR2). That is, people with MCI appear to experience a 10-year delay in AR1 compared to healthy people. Furthermore, it has been proven that the differences between healthy people and MCI people are already in AR1 (η^2^ = 0.28) and that task T3 is the only task that has had the strength to detect these differences (task 7 could possibly also have detected this difference; however, it is possible that the small sample size was responsible for the lack of test power). However, in this research, it was not possible to know at what age these differences begin to be significant. We could call this critical age the debut age in DCL. This aspect could be known through a cohort design.

Third, at the age of 70–79 years (AR2), individuals with MCI experience a marked decline in their functional capacity. Performance on tasks T1, T3, T4, T7, T10, T11, and T12 is much more impaired than the performance of healthy people over 80 years of age (AR3). That is, people with MCI appear to experience a delay greater than 10 years in AR2 compared to healthy people. It this age group in which the difference in performance between healthy people and MCI people is most remarkable (η^2^ = 0.43). Arguably, this age is a critical age not in the debut of MCI but in the flowering of MCI. From that age onwards, the decline in MCI people continues, but slowly.

Fourth, there is also a critical age for deterioration in task performance in healthy people. This critical age is undeniably observed in AR3. Just as in people with MCI, once significant decline occurs, decline continues, but it is possibly slower. However, in no case is this outbreak as strong as that experienced by people with MCI. In AR3, there are differences between Cases and Controls in tasks T1, T4, T5, T10, T11, and T12 (η^2^ = 0.28), but the magnitude of the differences between Cases and Controls is much smaller than that observed in AR2.

Fifth, the discriminant analysis reveals the same result as the MANOVA (replicating the result) and offers complementary information. It is enough to observe the value of the Cases and Controls centroids to appreciate, with the critical prudence that the good work of a scientist requires, that the hypotheses proposed as a result of the results discussed in the second, third, and fourth points above are plausible. In the set of 64 participants, the distances between the centroids of the Cases and Controls in AR1, AR2, and in AR3 are 0.554, 1.8, and 1.28, respectively.

To our knowledge, how healthy and MCI individuals behave regarding functional performance in IADLs according to age range has not been studied. Longitudinal studies have shown a progressive increase with age in limitations in complex daily functions in people with MCI [28,81] and people with prodromal signs of MCI [5]. These changes in everyday function are associated with cognitive processes such as processing speed, executive function, and memory, especially in financial tasks [82]. In this study, a discriminant analysis allows us to observe the functional performance of the PA-IADL of healthy people and people with MCI according to age range.

Having found that age is an imponderable variable that affects performance in tasks; having found that age is a moderating variable and, therefore, the differences in task performance between Cases and Controls are greater or lesser in the three arbitrarily formed ARs; and having found that some tasks are more affected than others in the different ARs and that the difference between Cases and Controls is not of the same magnitude in all of them, nor does it appear in the same ARs, it is worth asking what exists beyond, what exists regardless of age that explains these differences, and what cognitive abilities are involved in the tasks that explain these results? In the next section, we attempt to discuss this aspect.

### 4.3. Differences between Cases and Controls by Age Group and Type of Task: The Cognitive Demands Required by the Tasks Could Explain the Differences in Performance between Cases and Controls

It is plausible to hypothesize that the PA-IADL tasks that would best discriminate between Controls and Cases would be those that combine working memory, executive processes, and episodic memory [27,38,42,44]. Let us consider the three arbitrarily formed age ranges (60–69, 70–79, and 80 and over) and the type of task. We can see that in tasks T1, T4, T10, T11, and T12, the Cases have worse performance than the Controls from the age of 70. More cognitively demanding tasks are affected to a greater extent, such as task T1 (filling pill boxes) or task T10 (meal preparation), in which several cognitive processes are involved (cognitive flexibility, working memory, planning, and reasoning). Still, those in which only episodic memory is involved, such as task T12 (shopping for food), are affected to a lesser extent. Several studies have pointed out that people with MCI for whom several cognitive domains are affected perform worse in everyday tasks than people with MCI that who are affected in a single domain [7,83]. On the contrary, in tasks T3 (medication control) and T7 (prospective memory without cues), the deterioration pattern by age is different; it is very pronounced at AR1, but at advanced ages (80 and older), we found no differences between MCI and Healthy individuals; that is, at younger ages (60–70 years), MCI individuals have much worse performance than Healthy people or, in other words, the deterioration begins at an early age. These tasks are very cognitively demanding. In the case of task T7, the person must remember to deliver a document after 15 min without any help, and in task T3, the person must calculate for how many days there are pills left in a rheumatism bottle and to do so, they must perform up to six different well-planned steps (executive functioning) and different calculations (working memory). In this sense, we should not forget that when we perform an IADL, we set in motion different tasks simultaneously (in our case, steps) with several cognitive processes working together [6,84]. Thus, tasks T3 and T7 allow for the detection of people with MCI at an earlier age. This is very important for clinical practice since the earlier an MCI situation is detected, the earlier intervention can take place, and this will positively affect the evolution of cognitive functioning and the ability to perform daily activities autonomously. However, it is necessary to design new tasks of a similar nature to analyze convergent and predictive validity. Finally, in tasks T1 (filling pill boxes), T5 (bank direct debit), and T11 (bus route), the significant gap between MCI and Healthy individuals occurs at advanced ages (80 years and older). These tasks involve cognitive processes, such as working memory, planning, and cognitive flexibility, for which training could be provided to delay the deterioration in functional capacity as much as possible.

Once the eight tasks responsible for the most pronounced differences between Cases and Controls have been identified and the cognitive resources involved in them have been identified, could the score achieved in the eight tasks serve to identify who are Cases and who are Controls?

### 4.4. Convergence in the Classification of Cases and Controls of PA-IADL Test and Traditional (Standard) Cognitive Assessments

The five exploratory classification methods executed, with the predictor variables being performance in the eight most discriminative tasks, return the same result (strong replication). The five achieved a very high convergence in the classification of the participants with the traditional (standard) cognitive assessments, between 76.56% (K-means) and 81.24% (the hierarchical cluster procedure using the Ward method). The difference in means between the healthy people and people with MCI whom the two procedures identified is very relevant in each of the eight tasks.

Therefore, two things can be concluded. One is that the convergent validity of the assessment of the performance-based activities of daily living by the PA-IADL test with traditional (standard) cognitive assessments to identify older adults with mild cognitive impairment was demonstrated, and two, given that each of the eight tasks is useful for capturing the differences between Cases and Controls, given that in each task, there are several different cognitive abilities involved, and given that these cognitive abilities can be identified in the various “steps” into which each task is segmented, the identification of Cases and Controls using the PA-IADL test could be extremely useful to know which cognitive ability is most affected in a person, which cognitive ability deteriorates first, and which cognitive abilities are the most resilient and many more hypotheses that can be tested through appropriately planned experimental research to test the desired effect. It is also extremely useful because the alarm that “something is happening” in a person can be identified more quickly, even by a person’s close relatives, and early intervention can be carried out to stop or slow down the process. 

Another very striking result has to do with the performance in the tasks of the participants for whom there was no consensus on the classification, which we discuss in detail below.

On one hand, the six individuals classified as MCI by the k-means cluster method based on the PA-IADL and classified as Healthy by the cognitive assessment method have statistically equal performance in all tasks to those classified as MCI by both methods. These six people were over 80 years of age. Although they were cognitively well according to the cognitive tests, at that age, different processes are affected that have a decisive influence on daily functioning that the cognitive tests cannot detect. One might think that cognitive status is not a good predictor of daily functioning at that age or that the PA-IADL tasks are more accurate in discrimination because they involve a combination of different cognitive processes that are evaluated together in action and which, at that age, are more committed [85]. 

On the other hand, the nine people classified as Healthy by the k-means cluster method based on the PA-IADL and as MCI by the cognitive assessment have performance in all tasks which is statistically equal to those classified as Healthy by both methods. These nine people were younger, and although they performed worse in the cognitive tests than their normative group, this deterioration did not seem to affect the resolution of everyday tasks. That is to say, they presented fewer difficulties in solving daily tasks in which it is necessary to set in motion different cognitive processes simultaneously which are not so affected at these younger ages, due, perhaps, to less brain slowing.

From our point of view, these results invite us to hypothesize that the PA-IADL [45] test is more accurate than the traditional (standard) cognitive assessments in identifying people with MCI and Healthy people.

Traditionally, cognitive function is assessed through cognitive tests. One of these tests is the MMSE, which has problems in detecting mild cognitive impairment; specifically, a ceiling effect and lower sensitivity [86,87]. Our study used the MMSE to rule out people with moderate or severe cognitive impairment. We applied a broad battery of cognitive tests to analyze the relationship between cognitive processes and daily functioning, and this battery served, in turn, to detect people with mild cognitive impairment. Another test to evaluate cognitive function in older people is the MoCA. Some studies highlight the effectiveness of MoCA in detecting MCI [88,89]. However, in some MoCA tasks, older people must use decontextualized material and apply formal logic for which their personal experiences and stored knowledge are useless. In this type of cognitive assessment, it is easy to demonstrate losses associated with normal aging without pathology; hence, people without cognitive impairment could be classified as MCI. In contrast, assessing everyday functioning through performance-based tests provides insight into how older people cope with contextualized tasks, using logic based on everyday life and what these people do in their daily lives. 

Since cognitive processes are involved in everyday activities, the assessment of day-to-day functioning also makes it possible to analyze cognitive functioning and detect problems in this functioning from a more realistic perspective, avoiding issues in the classification of people with MCI and Healthy individuals. Thus, and in light of these results, it can be concluded that this is true.

### 4.5. Limitations 

This study has two important limitations. One is that the influence that the training or educational level has on the execution of the tasks was not evaluated. Different studies indicate that the training or academic level is a variable strongly related to cognitive functioning [90,91], and therefore, as has been evidenced in this research, also will be related to performance in the execution of tasks. However, this variable was used to match Cases and Controls, and therefore, the results found are free of systematic bias due to the training or educational level of the participants. Two, the small sample size. Although the data analysis carried out allowed us to conclude that each task behaves differently in the interaction (age range × EG Case/Control) and all the simple effects were statistically significant with a high effect size, we think that, perhaps, the small number of subjects did not allow us to find statistically significant differences in task 7 (prospective memory without cues) in AR1, so in future research, the sample size should be planned a priori [92], at least to test the most difficult hypotheses to test.

### 4.6. Summary of Most Relevant Contributions and Prospectives

However, despite these limitations, this study highlights the valuable contribution of the IADL-PA test as an assessment of functional capacity in the detection of individuals with MCI. On one hand, performance in the different tasks allows us to predict, regardless of age, whether a person is healthy or has MCI, which represents an added value of the PA-IADL to the extent that it will allow us to differentiate a deterioration in the processes and cognitive functions involved in daily tasks from a decline in cognitive functioning associated with age, especially in older adults. On the other hand, the set of functional tasks is capable of detecting MCI with greater precision because each of the tasks involves several cognitive competencies which, in combination, determine whether a task is performed successfully or not. Cognitive psychometric tests can never capture this aspect. Each of these cognitive tests participates in detection, as do all of them, but it is impossible to quantify what the interaction of several of them would mean, as in the case of functional tasks [93].

In addition to these two aspects that we consider to be of the utmost relevance, in the writing of this section dedicated to the discussion and conclusions, some new hypotheses were highlighted that arose in light of these results and raise new lines of research, some of which should be resolved through observational cohort research and others through experimental research.

## Figures and Tables

**Figure 1 behavsci-13-00975-f001:**
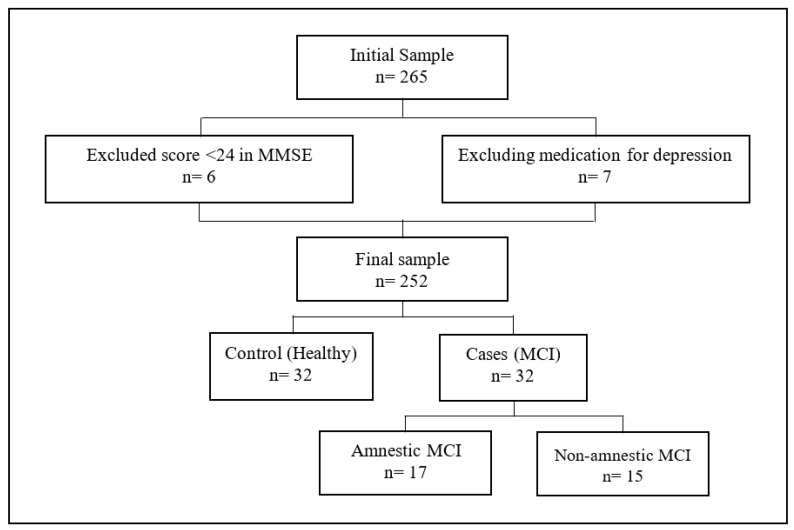
Sample selection.

**Figure 2 behavsci-13-00975-f002:**
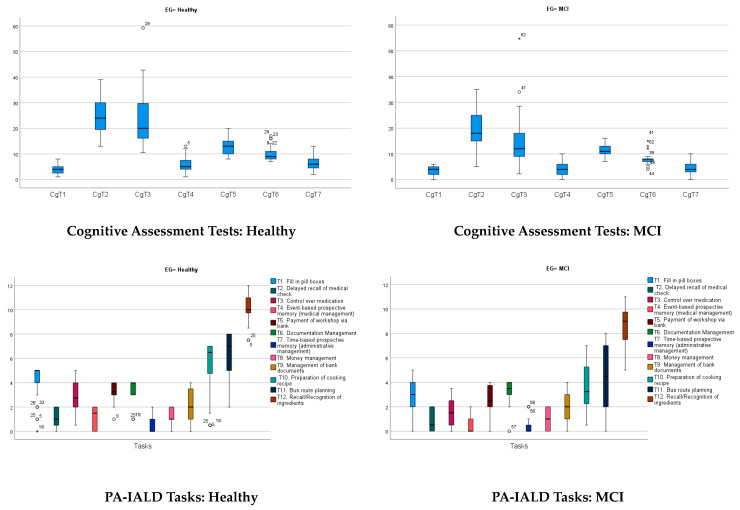
Top: graphical representation of the distribution of the seven tests used in the cognitive evaluation in each EG (◦ = outlier; * = extreme value). Bottom: graphical representation of the distribution of the 12 PA-IALD tasks in each EG.

**Figure 3 behavsci-13-00975-f003:**
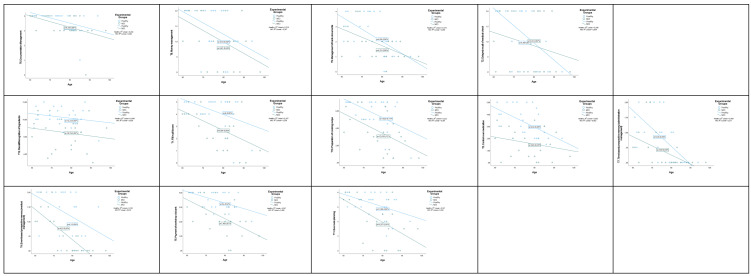
Differential performance in the 12 tasks of the PA-IALD in the groups of Cases (MCI) and Controls (Healthy) according to the age of the participants. Linear fit representation. *Legend*. The graphics are placed in order of shape. Tasks in **Row 1**: T6, T8, T9, and T2. Tasks in **Row 2**: T12, T1, T10, T3, and T7. Tasks in **Row 3**: T4, T5, and T11. See Appendix A, Section 2 (Graphics S2.G1, Graphics S2.G2, Graphics S2.G2 (continuation), and Graphics S2.G3) for an application of each of the graphics contained in Figure 3.

**Table 1 behavsci-13-00975-t001:** Mini-Mental State Examination (MMSE) and the six variables explored in the cognitive assessment (GgA). Descriptive statistics in each EG, the examination of differences between EGs, and a graphical representation of the MMSE distribution in each EG.

	M; SD; CV		^1^ F; *p*; η^2^	r_CA-AGE_		
Cognitive Assessment	Healthy	MCI		Healthy	MCI	Graphic Representation
MMSE	M = 29.47; SD = 0.879; CV = 0.029	M = 28.66; SD = 1.63; CV = 0.056	F = 6.11; *p* = 0.016; η^2^ = 0.090	−0.038	−0.112	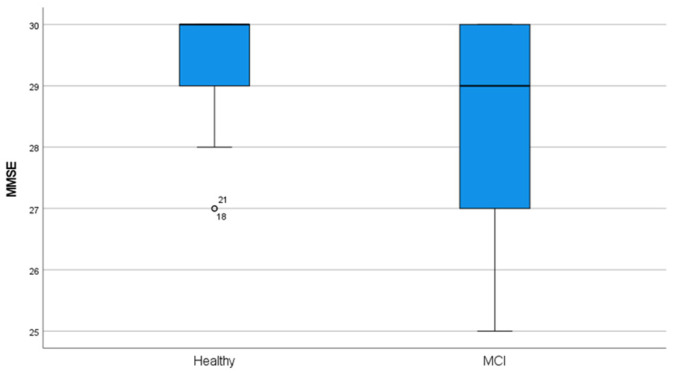
CgT1	M = 3.66; SD = 1.79; CV = 0.48	M = 3.06; SD= 1.93; CV = 0.63	F = 1.62; *p* = 0.207; η^2^ = 0.026	−0.468 **	−0.101	
CgT2	M = 24.31; SD = 6.50; CV = 0.26	M = 17.97; SD= 7.69; CV = 0.42	F = 12.70; *p* < 0.001; η^2^ = 0.170	−0.530 **	−0.151	
CgT3	M = 22.98; SD = 10.65; CV= 0.46	M = 14.28; SD= 10.18; CV = 0.71	F = 11.16; *p* = 0.001; η^2^ = 0.153	−0.703 **	−0.544 **	
CgT4	M = 5.75; SD= 2.82; CV = 0.49	M = 4.13; SD= 2.57; CV = 0.62	F = 5.80; *p* = 0.019; η^2^ = 0.086	−0.221	−0.169	
CgT5	M = 12.87; SD = 3.25; CV = 0.25	M = 11.32; SD= 2.44; CV = 0.21	F = 3.94; *p* = 0.053; η^2^ = 0.067	−0.242	−0.144	
CgT6	M = 9.94; SD= 2.96; CV= 0.29	M = 7.93; SD= 2.20; CV = 0.27	F = 9.23; *p* = 0.004; η^2^ = 0.131	−0.696 **	−0.285	
CgT7	M = 6.47; SD= 2.88; CV= 0.44	M = 4.50; SD= 2.41; CV = 0.53	F = 8.77; *p* = 0.004; η^2^ = 0.124	−0.572 **	−0.606 **	
Ys				−0.341	−0.428 *	

*Legend.* CgA = Cognitive assessment test (CgT1 = word list I (first attempt) (WMS-III); CgT2 = word list I (four attempt) (WMS-III); CgT3 = symbols and digits test; CgT4 = word list II (WMS-III); CgT5 = digit span (WAIS-III); CgT6 = arithmetic (WAIS-III); CgT7 = letter–number sequencing (WMS-III)); ^1^ = The variances between the EGs were heterogeneous only in the CgT6 evaluation (F_Levene_ = 4.12; gl_1_ = 1, gl_2_ = 60; *p* = 0.047), and therefore, to test the difference in measurements between the GAs in CgT6, the F_BF_ was used (in the rest the F of Fisher was used). In CgT5, there is 1 missing data point in EG Healthy and 7 missing data points in EG MCI, and in CgT6, there are 2 missing data points in MCI (in addition to heterogeneous variances); for this reason, *gl*_2_ = 54 and 57.05, respectively. In the rest, gl_2_ = 54. Always, *gl*_1_ = 1; r _CA-AGE_ = Pearson correlation between MMSE and CgA tests with age. * and ** = the correlation is significant at the 0.05 and 0.01 level, respectively. For to see a higher resolution graphical representation, see Appendix A (in the graphic, ◦ = outlier). For the rest, see Appendix A, and Figure 2 in the text.

**Table 2 behavsci-13-00975-t002:** Inferential results. A summary of the simple effects of the interaction (EG × AR: 2 × 3) in the MANOVA (dependent variables: T1, T3, T4, T5, T7, T10, T11, and T12) and a summary of the results of the discriminant analysis in 2 × 3 groups.

MANOVA ^A^									
	**Simple effects of the EG variable (Healthy-MCI) at each level of the AR variable (1 = 60–60, 2 = 70–79, and 3 = >80) and mean differences**								
	**Simple effects ^B^**	**^1^** **Mean differences**							
		**T1**	**T3**	**T4**	**T5**	**T7**	**T10**	**T11**	**T12**
**60–69 ^B1^**	Λ = 0.71; *p* = 0.022; η^2^ = 0.28	-----	DM = 1.82; *p* = 0.001	-----	-----	------	-----	-----	-----
**70–79 ^B2^**	Λ = 0.56; *p* = 0.000; η^2^ = 0.43	DM = 2.16; *p* = 0.003	DM = 2.30; *p* = 0.000	DM = 1.08; *p* = 0.001	-----	DM = 0.77; *p* = 0.012	DM = 3.38; *p* = 0.002	DM = 4.05; *p* = 0.003	DM = 2.19; *p* = 0.004
**>80 ^B3^**	Λ = 0.64; *p* = 0.001; η^2^ = 0.36	DM = 1.13; *p* = 0.021	-----	DM = 0.480; *p* = 0.023	DM = 1.11; *p* = 0.001		DM = 1.60; *p* = 0.028	DM = 2.05; *p* = 0.000	DM = 1.48; *p* = 0.004
	**Simple effects of the AR variable (1= 60–60, 2 = 70–79, and 3= >80) at each level of the EG variable (Healthy-MCI)) and ^1^ Mean Differences**								
	**Simple effects ^C^**	**^1^** **Mean differences**							
**Healthy ^C1^**	Λ = 0.52; *p* = 0.004; η^2^ = 0.28	-----	A_1–3_ = 1.80; *p* = 0.001 A_2–3_ = 1.65; *p* = 0.002	A_1–3_ = 1.18; *p* = 0.000 A_2–3_ = 0.700; *p* = 0.020	----	A_1–3_ = 1.12; *p* = 0.000 A_2–3_ = 0.81; *p* = 0.004	-----	-----	-----
**MCI ^C2^**	Λ = 0.40; *p* = 0.000; η^2^ = 0.37	A_1–2_ = 1.72; *p* = 0.054 A_1–3_ = 1.75; *p* = 0.007		A_1–2_ = 1.19; *p* = 0.001 A_1–3_ = 1.29; *p* = 0.000	A_2–3_ = 1.56; *p* = 0.000	A_1–3_ = 0.66; *p* = 0.019	A_1–2_ = 2.38; *p* = 0.084 A_1–3_ = 2.19; *p* = 0.032	A_1–2_ = 4.27; *p* = 0.000 A_1–3_ = 3.70; *p* = 0.000	
**Discriminant analysis**									
			**Group centroids**						
			**Healthy**				**MCI**		
**GV (6)** **EG × AR**	C. Standardized		**60–69**	**70–79**	**>80**		**60–69**	**70–79**	**>80**
	[T4 = 0.763; T11 = 0.679]	TS (n = 50)	1.721	1.435	−0.102		1.405	−1.888	−1.902
	[T4 = 0.783; T12 = 0.614]	Total sample (n = 64)	1.834	1.163	−0.206		1.380	−1.530	−1.487
	**Results of the discriminant analysis (inclusion by steps) in TS = 50 and ^D^VS = 14. Grouping variable: combination of levels of (EG × AR)**								
	TS (70% approx., N = 50) (Av = 2.559; %σ = 97.4; Rc = 0.848; Λ = 0.263 χ^2^ = 60.14; *gl* = 10; *p* = 0.000).								
	Correct classification in TS, 60%; correct classification in VS, 21.4%.								
	**Results of the discriminant analysis (inclusion by steps) in the total sample N = 64. Grouping variable: combination of levels (EG × AR)**								
	Total sample (N = 64) (Av = 1.87; %σ = 97.2; Rc = 0.807; Λ = 0.330; χ^2^ = 65.32; *gl* = 10; *p* = 0.000).								
	Correct classification in total sample, 50%.								

*Legend*. In MANOVA results (^1^ = Bonferroni correction was used; Λ = Wilks’ lambda test statistic; ^A^ = the empirical power of the statistical test (1−*β*) was always above 0.80; ^B^ = *df*_1_ and *df*_2_ are 8 and 51, respt., and ^C^ = *df*_1_ and *df*_2_ are 16 and 102, respt.; ^B1, B2, B3, C1 and-C2 =^ F statistic associated with Λ = 2.51; 4.94; 3.57; l2.45 and 3.71, respt.; AR = age range; A = age (subscripts 1, 2, and 3 indicate age ranges 60–69, 70–79, and >80, respt.); in discriminant analysis (GV = grouping variable used in the discriminant analysis (levels); C. Standardized = standardized coefficients of the relevant variables in the discrimination of the 6 VFs (EG × AR); TS and VS = training sample and validation sample, respt.; Av = eigenvalue; % σ = percentage of explained variance; Rc = canonical correlation; *df*; ^D^VS = 14: it was decided not to perform the discriminant analysis with the SV due to the small sample size). For the rest, see Figure 2.The study of the differences among means in the ARs reveals that in the age group 60–69 years, only the T3 task detected differences between Cases and Controls. Cases and Controls showed statistically significant differences in the age range 70–79 years in all tasks except T5 and in the age range > 80 years in all tasks except T3 and T7.

**Table 3 behavsci-13-00975-t003:** In the upper part, the graph of the distribution of the groups centroids of the six groups resulting from the interaction between the levels of the variables EG and AR in the solution of the discriminant analysis carried out in the total sample (n = 64). At the bottom, the force of the discriminating function (the introduction of variables by steps), through the statistic F, in the corresponding samples to discriminate between the 6 groups resulting from the interaction (EG × AR).

Total Sample N = 64						
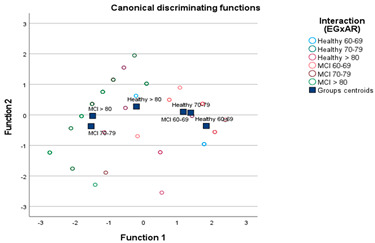						
	**Total sample. ^C^ Step 1: T4**					
	**H 60–69**	**H 70–79**	**H >80**	**MCI 60–69**	**MCI 70–79**	**MCI > 80**
**H 60–69**		2.795 (0.100)	20.849 (0.000)	1.649 (0.204)	24.056 (0.000)	43.364 (0.000)
**H 70–79**			7.921 (0.007)	0.160 (0.691)	12.142 (0.001)	23.799 (0.000)
**H >80**				10.635 (0.002)	1.810 (0.184)	5.415 (0.023)
**MCI 60–69**					14.761 (0.000)	28.466 (0.000)
**MCI 70–79**						0.135 (0.715)
	**Total sample. ^D^ Step 2: T12**					
	**H 60–69**	**H 70–79**	**H >80**	**MCI 60–69**	**MCI 70–79**	**MCI > 80**
**H 60–69**		1.378 (0.260)	11.697 (0.000)	0.818 (0.446)	19.061 (0.000)	29.777 (0.000)
**H 70–79**			5.265 (0.008)	0.105 (0.900)	13.222 (0.000)	20.369 (0.000)
**H >80**				7.062 (0.002)	4.561 (0.015)	6.793 (0.002)
**MCI 60–69**					15.319 (0.000)	23.800 (0.000)
**MCI 70–79**						0.254 (0.777)

*Legend*. ^C,D^ = degrees of freedom in steps 1 and 2 of the statistic F (1:58 and 2:57, respt.); H = Healthy.

**Table 4 behavsci-13-00975-t004:** Percentage of coincidence and Kappa coefficient between the five cluster analysis procedures examined and the classification made using the cognitive tests (starting point) in the total sample (N = 64).

Classification Methods	Cl.Cg	
	^1^ %Mc	Kappa
J_W.M_	81.24 (12)	0.625 (*p* = 0.000)
J_FN.M_	79.68 (13)	0.594 (*p* = 0.000)
Two-Stage	79.68 (13)	0.594 (*p* = 0.000)
Discriminant	79.68 (13)	0.594 (*p* = 0.000)
K-Means	76.56 (15)	0.531 (*p* = 0.000)

*Legend*. M = method; J_W.M_, J_FN.M_ = classification carried out via the hierarchical cluster procedure using the Ward method and the farthest neighbor method, respectively; Cl.Cg = classification performed using cognitive tests (traditional classification mode. Starting point); %Mc = percentage of matching cases in the classification; ^1^ = the number of mismatched cases is shown in parentheses; *p* = *p* value.

**Table 5 behavsci-13-00975-t005:** Descriptive statistics (M and SD) in the 8 target tasks in each of the groups resulting from crossing the classification of the Case (MCI) and Control (Healthy) participants between the traditional cognitive tests and the results derived from the cluster analysis procedures, hierarchical (Ward’s method) and K-Means (greater and lesser coincidence in classification), and the analysis of differences between the means of the 4 groups that make up the confusion matrix.

										^A^ ANOVA_BF_
**Confusion Matrix**	**Cl.Cg × J_W.M_ ^1^ [81.24% (12)]**									^B^ F_BF_T1 = 19.41; *gl*_2_ = 16.68; *p* = 0.00
**Cl.Cg × J_W.M_**	**N**	**T1**	**T3**	**T4**	**T5**	**T7**	**T10**	**T11**	**T12**	^B^ F_BF_T3 = 12.86; *gl*_2_ = 21.85; *p* = 0.00
Healthy_C_- Healthy_P_	**26**	4.58 (0.70)	3.28 (1.28)	1.23 (0.86)	3.46 (0.73)	0.75 (0.82)	6.26 (1.13)	6.77 (1.24)	10.40 (0.89)	^B^ F_BF_T4 = 12.06; *gl*_2_ = 17.56; *p* = 0.00
MCI_C_-MCI_P_	**26**	2.62 (1.44)	1.42 (1.01)	0.27 (0.40)	2.23 (1.12)	0.15 (0.30)	2.92 (1.70)	3.58 (2.64)	8.23 (1.58)	^B^ F_BF_T5 = 12.81; *gl*_2_ = 32.11; *p* = 0.00
Healthy_C_-MCI_P_	**6**	1.83 (1.47)	1.66 (0.75)	0.67 (0.87)	2.75 (0.41)	0.17 (0.40)	1.75 (1.60)	4.00 (1.41)	9.08 (1.39)	^B^ F_BF_T7 = 5.50; *gl*_2_ = 16.98; *p* = 0.00
MCI_C_-Healthy_P_	**6**	4.50 (0.54)	2.50 (1.34)	1.67 (0.40)	3.67 (0.81)	1.00 (0.83)	6.58 (0.80)	7.67 (0.51)	10.17 (0.81)	^B^ F_BF_T10 = 40.29; *gl*_2_ = 22.43; *p* = 0.00
										^B^ F_BF_T11 = 24.89; *gl*_2_ = 36.43; *p* = 0.00
										^B^ F_BF_T12 = 15.21; *gl*_2_ = 23.69; *p* = 0.00
	**Cl.Cg × K-Means ^1^ [76.56% (15)]**									^B^ F_BF_T1 = 22.41; *gl*_2_ = 16.17; *p* = 0.00
**Cl.Cg × K-Means**		**T1**	**T3**	**T4**	**T5**	**T7**	**T10**	**T11**	**T12**	^B^ F_BF_T3 = 14.45; *gl*_2_ = 41.74; *p* = 0.00
Healthy_C_-Healthy_P_	**26**	4.58 (0.70)	3.29 (1.28)	1.23 (0.86)	3.46 (0.73)	0.75 (0.82)	6.27 (1.13)	6.77 (1.24)	10.40 (0.89)	^B^ F_BF_T4 = 9.78; *gl*_2_ = 20.61; *p* = 0.00
MCI_C_-MCI_P_	**23**	2.39 (1.37)	1.37 (1.06)	0.22 (0.39)	2.22 (1.02)	0.15 (0.31)	2.83 (1.77)	3.13 (2.47)	8.13 (1.64)	^B^ F_BF_T5 = 7.09; *gl*_2_ = 20.96; *p* = 0.00
Healthy_C_-MCI_P_	**6**	1.83 (1.47)	1.67 (0.75)	0.67 (0.88)	2.75 (0.41)	0.17 (0.40)	1.75 (1.60)	4.00 (1.41)	9.08 (1.39)	^B^ F_BF_T7 = 4.70; *gl*_2_= 28.15; *p* = 0.00
MCI_C_-Healthy_P_	**9**	4.44 (0.52)	2.28 (1.12)	1.33 (0.61)	3.22 (1.39)	0.72 (0.79)	5.61 (1.64)	7.44 (0.52)	9.78 (0.97)	^B^ F_BF_T10 = 27.29; *gl*_2_ = 28.06; *p* = 0.00
										^B^ F_BF_T11 = 24.89; *gl*_2_ = 36.43; *p* = 0.00
										^B^ F_BF_T12 = 13.65; *gl*_2_ = 26.89; *p* = 0.00

*Legend*. Confusion matrix (four categories: Healthy_C_-Healthy_P_ = participants that both approaches, the PA-IALD target tasks and the traditional cognitive tests, coincide in classifying as healthy people; MCI_C_-MCI_P_ = participants that both approaches agree in classifying as MCI people; Healthy_C_-MCI_P_ = participants that cognitive tests classify as healthy people and that the PA-IALD target tasks classify as MCI people; MCI_C_-Healthy_P_ = participants that the cognitive tests classify as MCI people and that the PA-IALD target tasks classify as healthy people); ^1^ = percentage of cases coincident in the classification (number of non-coinciding cases); ANOVA_B F_= was evaluated using F_BF_); ^A^ = *df*_1_ = 3 always; *gl*_2_ = degrees of freedom of the error associated with the F_BF_; ^B^ = the mean comparison analysis revealed that there are statistically significant differences between Healthy_C_-Healthy_P_ and MCI_C_-MCI_P_, but there are no differences between the groups Healthy_C_-Healthy_P_ and MCI_C_-Healthy_P_, nor between the groups MCI_C_-MCI_P_ and Healthy_C_-MCI_P_.

## Data Availability

The data presented in this study are available upon request from the corresponding authors.

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
