# Peer review of "Analysis of Convergent Validity of Performance-Based Activities of Daily Living Assessed by PA-IADL Test in Relation to Traditional (Standard) Cognitive Assessment to Identify Older Adults with Mild Cognitive Impairment"

_behavsci, 2023, doi:10.3390/bs13120975_

Round 1

Reviewer 1 Report

Comments and Suggestions for Authors

Dear authors,

I congratulate you on your work. Here are a some minor revisions that need to be made before publication, mostly related to the presentation of the results:

- In the abstract, "mild cognitive impairment" should be added before the acronym MCI. The same applies to line 34, where the acronym IADL should be explained.

- Images and graphs included in the tables should be extracted from the and removed. A lot of information is provided, so it is recommended to include this information only as complementary material.

- Data analysis section should be summarised and clarified.

- Table 6 should appear on a separate page, horizontally, like table 1.

I hope my suggestions and indications will be useful.

Best regards,

Author Response

Response to Reviewer 1 Comments

We want to thank the reviewer for the careful and critical reading of this article because their considerations have contributed to substantially improving the research work presented in this manuscript, and the presentation of the manuscript itself.

In response to the reviewer's requests, a large part of the article has been modified; without a doubt, it is now much clearer, more informative, and easier to read, and the motivation and development of the proposed procedure are much better understood.

We respond in detail to each of your wise suggestions as follows.

-We highlight in black as Point 1, Point 2, etc., the suggestions made by the reviewer.

-We highlight the answers to each of the previous points in red. As follows: Response for Point 1, Response for Point 2, etc.

-In the new version of the scientific article, all the changes made have been introduced and explained. The new text that has been added has been highlighted in blue so that the changes made can be seen more clearly.

-In this response document to the reviewer, some parts of the paper are shown literally. These portions of text that are displayed verbatim are displayed tabulated. The reviewer can verify that they appear identically in the new version of the paper.

Comments and Suggestions for Authors

I congratulate you on your work. Here are a some minor revisions that need to be made before publication, mostly related to the presentation of the results:

Point 1. In the abstract, "mild cognitive impairment" should be added before the acronym MCI. The same applies to line 34, where the acronym IADL should be explained.

Response for Point 1: The reviewer is right:

-In the abstract, the expression "mild cognitive impairment” has been added before the acronym MCI, which has been introduced in parentheses.

-In the first paragraph of the introduction, the expression “instrumental activities of daily living” is added before the acronym IADL, and this is introduced in parentheses.

-We have considered it appropriate to change the article's title, replacing the acronym MCI with the expression “mild cognitive impairment.” As follows:

Analysis of convergent validity of performance-based activities of daily living assessed by PA-IADL test in relation to traditional (standard) cognitive assessment to identify older adults with mild cognitive impairment.

Point 2. Images and graphs included in the tables should be extracted from the and removed. A lot of information is provided, so it is recommended to include this information only as complementary material

Response for Point 2. We agree with the reviewer that it can be too much information.

However, we consider that, in this specific case, the graphs introduced in Tables 1, 2, and 4 are relevant because, at a glance, you can see in the graphs of Tables 1 and 2 the differences between the two subgroups, Cases (or MCI) and Controls (or Healthy individuals), and in Table 4, the distribution of the Groups Centroids of the 6 groups resulting from the interaction between the levels of the variables EG and AR in the solution of the Discriminant analysis carried out in the total sample (n = 64).

Considering this aspect, we have decided to keep the graphs of Tables 1, 2, and 4 in the text. However, we are very grateful to the reviewer for this request because, due to it, we have reviewed all the tables and have made several changes. The following:

-We have placed the layout of two tables (Table 2 and Table 6, the latter, at the reviewer's request, shown in Point 4, which we consider very accurate).

-We have identified some errors that we have corrected. Specifically, we have changed the following aspects:

-We have slightly corrected the legend of Table 1.

-We have placed Table 2 on a single-page

-We have slightly modified Table 5 to present it more compactly.

-We have placed Table 6 in landscape mode on a single page (in response to the reviewer's Point 4)

-We have replaced in the Table 3 with the centroid value in MCI -.1888 with 1.888.

Point 3. - Data analysis section should be summarised and clarified.

Response for Point 3. The reviewer is right.

Demonstrating the convergent validity of both modes of MCI diagnosis requires a complex network of linked, properly sequenced data analysis that can be difficult to follow. We have reviewed the entire section and decided to reorder the exhibition. We have also introduced some added explanations. However, we have not eliminated anything from the text contained in this section since all of this is expressly necessary to understand the reasoning we have followed.

The text of section 2.3. Data Analysis now displays as:

2.3. Data analysis

Each of the two previously written objectives (see section Introduction) implies subordinate objectives that must be carried out in a certain order, and the analysis of the data is conditioned by the activities involved in them. The detailed objectives are as follows. First, the two objectives are described in detail, and then the data analysis sequence involved in carrying out each objective is defined.

Objective 1: Study of the convergent validity of PA-IADL functional tasks with cognitive assessment tests (traditional MCI detection method), hereinafter CgA, in their capacity as MCI detection tests. This contains two subordinate objectives:

Subordinate objective 1.1- To determine which PA-IADL tasks are useful to identify and differentiate participants who have been classified as MCI and participants who have been classified as Healthy by CgA tests. Because all PA-IADL tasks require cognitive strategies for their performance and cognitive skills are naturally affected by age, to achieve this subordinate objective it was considered necessary to determine two aspects. First, we assessed which tasks are useful for discriminating between MCI and Healthy individuals regardless of the age of the participants. Second, we assessed whether there were differences between MCI and Healthy participants in their performance according to age. The result of these analyses will allow us to understand which tasks have the greatest power to discriminate between MCI participants and Healthy ones, and we will call these target tasks (TT).

Subordinate objective 1.2- To evaluate the effectiveness of the TT set in classifying the participants of the sample set into clearly differentiated categories.

Objective 2: To analyze the divergences in the MCI-Healthy classification of the two methods, through the functional tasks of the PA-IADL (only the TTs) and through the CgA, in the diagnosis of MCI. This contains two subordinate objectives:

Subordinate objective 2.1- To examine the differences in the performance of the TTs of the PA-IADL between the participants that make up the 4 groups of the confusion matrix, with particular interest in the participants that the two methods classify in different groups.

Subordinate objective 2.2- To examine what characteristics are shown by the participants that the two methods classify in different groups in the variables used to carry out the matching.

The data analysis carried out to carry out each of the two subordinate objectives contained in Objective 1 and Objective 2 was as follows.

2.3.1. Data analysis in pursuit of subordinate objective 1.1

At this point, it is a priority to understand whether it is possible to study the 12 tasks together, or on the contrary, this must be done individually. The statistical advantage of multivariate tests over univariate tests is that they keep Type I error under control (Hair et al., 2006; Maxwell et al., 2017). But there is also a substantive advantage that in this particular case is very relevant, and that is that they would allow a holistic study to be carried out of the integrated PA-IADL package, in the same way that cognitive tests in the diagnosis of MCI constitute a holistic package. To do this, we began by evaluating the correlation in the performance of the 12 tasks with each other (see Table S3 in the supplementary material), and it was found that the 12 tasks are intercorrelated. Specifically, our findings were as follows: 89.39% (n=59) of the correlations were statistically significant; 45% (n=30) of the correlations had a magnitude greater than .40, the minimum value for applying a multivariate methodology (Cochran & Cox, 1957); no correlation reached the value of .80, therefore it is unlikely that there is multicollinearity (Bray & Maxwell, 1985; Tabachnick & Fidel, 1989); and Bartlett's sphericity test allows us to reject the null hypothesis that the residual covariance matrix is proportional to an identity matrix (χ2= 371.42; df=77; p=.001). For all these reasons, it is concluded that it is more appropriate to use a multivariate methodology than a univariate one (Hair et al., 2007; Maxwell & Delaney, 2004).

Thus, it was decided to carry out first a MANCOVA taking age as a covariate and then a MANOVA (EG x Age groups) introducing age as a classification variable. The two analyses provide us with complementary information. We will outline this below briefly.

Given that age has a statistically significant linear relationship with all tasks (except task 12) (see Table S3 in supplementary material), it is a priority to determine the tasks in which the difference in performance between Cases and Controls disappears when controlling for the effect of age, and the tasks in which the differences between the two groups exist even when controlling for the effect of age. This is the reason that justifies the MANCOVA, and we began by checking whether the rest of the assumptions it requires were satisfied. It was verified that there was no relationship between the covariate and the assigned independent variable (Eg), and it was also confirmed that Tasks 3 and 7 show non-parallel slopes in the groups of Cases and Controls. Based on this result, Task 12 was examined using ANOVA; for the study of Tasks 3 and 7, the regions of non-significance and significance were determined using the Johnson-Neyman technique (see Huitema, 2011) and performing the ANCOVA in the regions of significance; and the joint analysis of the remaining 10 tasks was performed using the MANCOVA following the modeling strategy proposed by Milliken & Johnson (2001). Since the difference in the execution of Tasks T2, T6, T8, and T9 between the EG disappears once the effect of age is controlled, these four tasks did not form part of the subsequent MANOVA.

Because the sample of participants covers ages between 60 and 98 years, for practical purposes, in addition to understanding whether age influences the performance of the different tasks, it is of great interest to understand whether the performance in the groups of Cases and Controls is moderated by age. To determine this, the sample was segmented into three age ranges (AR) (60-69, 70-79, and over 80 years), and a MANOVA (2x3) (EG x AR) was performed on the set of 8 tasks in which, once the effect of age was controlled, the differences between the EGs persisted. The result was examined using the Wilks Lambda statistic because it is the most robust test when the assumption of homogeneity of the covariance matrices is not satisfied (see Ates et al., 2019), as in this case [M de Box =143.53; g1=78 and 12138.7; p=.005; Levene's test detected non-homogeneous variances in tasks 4, 5, 11, and 12]. Since the interaction was statistically significant, we proceeded to study the simple effects and their corresponding mean differences (Bonferroni correction). The analysis was completed by performing a stepwise discriminant analysis to examine which combination of dependent variables had the strongest power to differentiate between the 6 groups resulting from the combination (EG x AR), paying special attention to the standardized coefficients and the magnitude and sign of the centroids. The analysis was first performed on a randomly chosen training sample (70% of cases, approx. n=50), and then on the whole sample. (It was decided not to perform the analysis on the validation sample, because 30 % of the sample, the remaining sample, is too small to be useful as a validation sample).

The reference values are the p-value (α=.05), partial eta squared (η2) (.01 small, .06 medium, and .14 large. See Cohen, 2013, Ellis, 2010), and (1-β) >.80.

To aid in the interpretation of the results obtained by both MANCOVA and MANOVA, the linear relationship between task performance and age was plotted, and also, the line of fit was calculated by iterative weighted least squares (Loess, using the Epanechnikov kernel function, where data located near a particular point receive higher weights than data that are further away).

2.3.2. Data analysis in the achievement of the subordinate objective 1.2

The previous analyses led to the conclusion that there were 8 TTs. Because the classification techniques are exploratory techniques and due to the small sample size, it was considered necessary to control the potential overfitting that could occur, and also to control the effect of the method used for classification. Thus, the analysis was carried out using 5 different methods, the hierarchical cluster procedure using two methods—the furthest neighbor method, and the Wald method— (both were chosen because they are robust to outlier scores, a probable case in such a small sample), the two-stage cluster method, the k-means method, and the discriminant method. The last two require the indicating of the number of clusters, the hierarchical procedures do not, and it is optional in the two-stage cluster analysis (in this case, it was performed without determining the number of clusters). Furthermore, the analysis was also performed by cross-validation, randomly selecting a new training sample (approximately 70% of cases, n=42). Again, due to the small size of the validation sample (n=22), the second analysis was performed on the full sample.

The percentage of coincidence was calculated between the MCI-Healthy classifications (i.e., Cases-Controls) carried out with the 5 classification techniques performed based on the TTs and the original classification carried out using the CgA tests. The magnitude of the agreement was also calculated using the Kappa coefficient.

Objective 2: To analyze the divergences in the MCI-Healthy classification of the two methods, through the functional tasks of the PA-IADL (only the TTs) and through the CgA, in the diagnosis of MCI. This contains two subordinate objectives:

Subordinate objective 2.1- To examine the differences in the performance of the TTs of the PA-IADL between the participants that make up the 4 groups of the confusion matrix, with particular interest in the participants that the two methods classify in different groups.

Subordinate objective 2.2- To examine what characteristics are shown by the participants that the two methods classify in different groups in the variables used to carry out the matching.

2.3.3. Data analysis in the achievement of subordinate objective 2.1

The decision was made to take as a reference the classification resulting from the hierarchical analysis using the Ward method and the resulting classification using the k-means method, as these were the classifications that had the highest and lowest concordance between the two methods. The differences between the 4 groups of the confusion matrix in the 8 TTs were tested using the robust Brown-Forsythe ANOVA due to the heterogeneity of the variances between the groups, in part, caused by the different size of the groups of the confusion matrix (Vallejo et al., 2010).

2.3.4. Data analysis in the achievement of subordinate objective 2.2

This was done by looking at the descriptive statistics. The analysis of the two subordinate objectives will allow us to identify the power, limitations, and vulnerability limits of the detection procedure based on cognitive tests.

The most crucial thing in a case-control study is to choose “the cases” carefully and “the controls”. If the selection is not correct, and if the matching is not accurate, there will be selection bias, and the inferences will not be valid. This is one of the reasons that justifies this section being so extensive and so detailed. Much effort was invested to correctly identify cases and controls, with the sole objective that the cases were indisputably cases and that the controls were indisputably controls. For this reason, the sample size was only 64 people in total. On the other hand, the data analysis that was carried out was in-depth and very orderly. As can be seen, each objective was carried out with different analyses as a “sensitivity analysis” to qualify the results and safeguard their replicability.

All analyses were performed using the statistical package IBM SPSS 27.

Point 4.- Table 6 should appear on a separate page, horizontally, like table 1.

Response for Point 4. We agree with the reviewer.

As explained in the response made in point 2, Table 6 has been presented in landscape mode.

Reviewer 2 Report

Comments and Suggestions for Authors

The aim of this study is very interesting, because focus on the importance of the IADL assessment as a complement to the cognitive assessment and mood for the detection of MCI, which is especially important due to the progressive growth of the elderly population.

However, some suggestions are indicated below in order to improve some sections of the manuscript:

- Cognitive function is assessed by MMSE, but it should also be interesting considering MoCa, that is a screening test to detect MCI.

- The size of the sample could be larger, especially participants with MCI, in other to be more representative of the general population, although the results are very interesting and can be useful for the clinical practice.

- The authors already take into consideration that they have not considered the variable of educational level, which is fundamental in the neuropsychological evaluation, but this has been already included as a limitation.

- The structure and content of the paper are very good and clear, however the length of some sections should be considered, since they are probably very long, especially the procedure. It could probably be synthesized and some of the content could be included in the results.

- The references are very competed, but we suggested to replace some of the oldest if possible with more recent ones from the last 3 years, to better update recent studies in the literature regarding MCI and the importance of its early detection.

Author Response

Response to Reviewer 2 Comments

We want to thank the reviewer for the careful and critical reading of this article because their considerations have contributed to substantially improving the research work presented in this manuscript, and the presentation of the manuscript itself.

In response to the reviewer's requests, a large part of the article has been modified; without a doubt, it is now much clearer, more informative, and easier to read, and the motivation and development of the proposed procedure are much better understood.

We respond in detail to each of your wise suggestions as follows.

-We highlight in black as Point 1, Point 2, etc., the suggestions made by the reviewer.

-We highlight the answers to each of the previous points in red. As follows: Response for Point 1, Response for Point 2, etc.

-In the new version of the scientific article, all the changes made have been introduced and explained. The new text that has been added has been highlighted in blue so that the changes made can be seen more clearly.

-In this response document to the reviewer, some parts of the paper are shown literally. These portions of text that are displayed verbatim are displayed tabulated. The reviewer can verify that they appear identically in the new version of the paper.

Comments and Suggestions for Authors

The aim of this study is very interesting, because focus on the importance of the IADL assessment as a complement to the cognitive assessment and mood for the detection of MCI, which is especially important due to the progressive growth of the elderly population.

However, some suggestions are indicated below in order to improve some sections of the manuscript:

Point 1. - Cognitive function is assessed by MMSE, but it should also be interesting considering MoCa, that is a screening test to detect MCI.

Response for Point 1: The reviewer is right.

Section 4. Discussion has been completely modified. It has been redrafted to present the discussion of all the results found and the conclusions derived with greater rigor and clarity. This section is now called 4. Discussion and conclusions and it responds to the reviewer's very correct suggestion, as follows (highlighted in yellow):

4.4. Convergence in the classification of Cases and Controls of PA-IADL test and traditional (standard) cognitive assessment.

The five exploratory classification methods executed, with the predictor variables being performance in the eight most discriminative tasks, return the same result (strong replication). The 5 achieved a very high convergence in the classification of the participants with the traditional (standard) cognitive assessment, between 76.56% (K-means) and 81.24% (Hierarchical Cluster procedure using the Ward method). The difference in means between the healthy people and people with MCI whom the two procedures have identified is very relevant in each of the eight tasks.

Therefore, two things can be concluded. One that the convergent validity of performance-based activities of daily living assessed by PA-IADL test about traditional (standard) cognitive assessment to identify older adults with mild cognitive impairment has been demonstrated, and two, given that each of the eight tasks is useful to capture the differences between Cases and Controls, given that in each task there are several different cognitive abilities involved, given that these cognitive abilities can be identified in the various “steps” into which each task is segmented, the identification of Cases and Controls using PA-IADL could be extremely useful to know which cognitive ability is most affected in a person, which cognitive ability deteriorates first, which cognitive abilities are most resilient, and much more of hypotheses that can be tested through appropriately planned experimental research to test the desired effect. It is also extremely useful because the alarm that “something is happening” in a person can be identified more quickly, even by the person's close relatives, and early intervention can be carried out to stop or slow down the process.

Another very striking result has to do with the performance in the tasks of the participants in which there was no consensus in the classification. In detail:

On the one hand, the six individuals classified as MCI by the k-means cluster method based on the PA-IADL, and classified as Healthy by the cognitive assessment method have statistically equal performance in all tasks to those classified as MCI by both methods. These six people were over 80 years of age. Although they were cognitively well according to the cognitive tests, however at that age, different processes are affected that have a decisive influence on daily functioning that the cognitive tests cannot detect. One might think that cognitive status is not a good predictor of daily functioning at that age or that the PA-IADL tasks are more accurate in discrimination because they involve the combination of different cognitive processes that are evaluated together in action and that at that age are more committed [85].

On the other hand, the nine people classified as Healthy by the k-means cluster method based on the PA-IADL and as MCI by the cognitive assessment have a performance in all tasks statistically equal to those classified as Healthy by both methods. These nine people were younger, and although they performed worse in the cognitive tests than their normative group, this deterioration did not seem to affect the resolution of everyday tasks. That is to say, they presented fewer difficulties in solving daily tasks in which it is necessary to set in motion different cognitive processes simultaneously and which, at these younger ages, are not so affected due, perhaps, to less brain slowing.

From our point of view, these results invite us to hypothesize that PA-IADL [85] is more accurate than the traditional (standard) cognitive assessment to identify people with MCI and Healthy people.

Traditionally, cognitive function is assessed through cognitive tests. One of these tests is the MMSE, which has problems in detecting mild cognitive impairment, specifically, a ceiling effect and lower sensitivity [86, 87].  Our study used the MMSE to rule out people with moderate or severe cognitive impairment. We applied a broad battery of cognitive tests to analyze the relationship between cognitive processes and daily functioning, and this battery served, in turn, to detect people with mild cognitive impairment. Another test to evaluate cognitive function in older people is the MoCA. Some studies highlight the effectiveness of MoCA in detecting MCI [88, 89]. However, in some MoCA tasks, older people must use decontextualized material and apply formal logic, where their personal experiences and stored knowledge are useless. In this type of cognitive assessment, it is easy to demonstrate the losses associated with normal aging without pathology; hence, people without cognitive impairment could be classified as MCI. In contrast, assessing everyday functioning through performance-based tests provides insight into how older people cope with contextualized tasks, with a logic based on the every day and what these people do in their daily lives.

Since cognitive processes are involved in everyday activities, the assessment of day-to-day functioning also makes it possible to analyze cognitive functioning and detect problems in this functioning from a more realistic perspective, avoiding issues in the classification of people with MCI and Healthy individuals. Thus, and in light of these results, it could be concluded that this is true.

Point 2. The size of the sample could be larger, especially participants with MCI, in other to be more representative of the general population, although the results are very interesting and can be useful for the clinical practice.

Response for Point 2. The reviewer is right that it would be desirable to have had a larger sample size. However, it was not possible.

As stated in the title of the article, the purpose of the research carried out was:

Analysis of convergent validity of performance-based activities of daily living assessed by PA-IADL test in relation to traditional (standard) cognitive assessment to identify older adults with mild cognitive impairment.

To do this, it was carried out, as indicated in the final part of the introduction:

a cross-sectional study of case (MCI) - control (Healthy) with a causal-comparative and predictive purpose with two main objectives:

The most crucial thing in a case-control study is to choose the “cases carefully” and “the controls” (Schulz and Grimes, 2002). If the selection is incorrect, firstly, and secondly, if the matching is not accurate, there will be selection bias, and the inferences will not be correct.

For this reason, section 2 of this article is long and detailed. A lot of effort and time was invested in identifying cases and controls. We wanted the cases to be cases indisputably and the controls to be controls indisputably. For this reason, the sample size was finally made up of 64 people.

The data analysis that was carried out was in-depth and very orderly. As can be seen, each hypothesis was tested with different analyses as a “sensitivity analysis” to safeguard the replicability of the results. This aspect is described and added at the end of section 2.3. Data analysis, as follows:

The most crucial thing in a case-control study is to choose “the cases” carefully and “the controls”. If the selection is not correct, and if the matching is not accurate, there will be selection bias, and the inferences will not be valid. This is one of the reasons that justifies this section being so extensive and so detailed. Much effort was invested to correctly identify cases and controls, with the sole objective that the cases were indisputably cases and that the controls were indisputably controls. For this reason, the sample size was only 64 people in total. On the other hand, the data analysis that was carried out was in-depth and very orderly. As can be seen, each objective was carried out with different analyses as a “sensitivity analysis” to qualify the results and safeguard their replicability.

Schulz, K. F., & Grimes, D. A. (2002). Case-control studies: research in reverse. The lancet359(9304), 431-434.

This aspect regarding the small size of the sample had also been highlighted in the limitations indicated in the Discussion section (4. Discussion and conclusions), as follows:

4.5. Limitations

This study has two important limitations. One, the influence that the training or educational level has on the execution of the tasks has not been evaluated. Different studies indicate that the training or academic level is a variable strongly related to cognitive functioning [90, 91], and therefore, as has been evidenced in this research, also will be related to performance in the execution of tasks. However, this variable was used to match Cases and Controls, and therefore, the results found are free of systematic bias due to the training or educational level of the participants. Two, the small sample size. Although the data analysis carried out has allowed us to conclude that each task behaves differently in the interaction (Age Range x EG Case/Control), and all the simple effects have been statistically significant with a high effect size, we think that, perhaps, the small number of subjects does not allow us to find statistically significant differences in task 7 (prospective memory without cues) in AR1, so in future research the sample size should be planned a priori [92], at least to test the most difficult hypotheses to test.

Point 3. The authors already take into consideration that they have not considered the variable of educational level, which is fundamental in the neuropsychological evaluation, but this has been already included as a limitation.

Response for Point 3. The reviewer is right. This aspect and the one indicated in Point 2 are noted in the limitations, as follows:

4.5. Limitations

This study has two important limitations. One, the influence that the training or educational level has on the execution of the tasks has not been evaluated. Different studies indicate that the training or academic level is a variable strongly related to cognitive functioning [90, 91], and therefore, as has been evidenced in this research, also will be related to performance in the execution of tasks. However, this variable was used to match Cases and Controls, and therefore, the results found are free of systematic bias due to the training or educational level of the participants. Two, the small sample size. Although the data analysis carried out has allowed us to conclude that each task behaves differently in the interaction (Age Range x EG Case/Control), and all the simple effects have been statistically significant with a high effect size, we think that, perhaps, the small number of subjects does not allow us to find statistically significant differences in task 7 (prospective memory without cues) in AR1, so in future research the sample size should be planned a priori [92], at least to test the most difficult hypotheses to test.

Point 4. The structure and content of the paper are very good and clear, however the length of some sections should be considered, since they are probably very long, especially the procedure. It could probably be synthesized and some of the content could be included in the results.

Response for Point 4. The reviewer is right. As indicated in the Response for Point 2, we have added a small paragraph to justify the extension of section 2, Materials and Methods. Consequently, the extension of section 3, Results, is also justified.

However, since demonstrating the convergent validity of both modes of diagnosis of the MCI requires a complex network of chained and properly sequenced data analysis, which can be difficult to follow, we have reviewed the entire Data analysis section (section 2.3), and we have decided to rearrange the exhibition. We have also introduced some added explanations. However, we have not eliminated anything from the text contained in this section since all of this is expressly necessary to understand the reasoning we have followed.

The text of section 2.3. Data Analysis now displays as:

2.3. Data analysis

Each of the two previously written objectives (see section Introduction) implies subordinate objectives that must be carried out in a certain order, and the analysis of the data is conditioned by the activities involved in them. The detailed objectives are as follows. First, the two objectives are described in detail, and then the data analysis sequence involved in carrying out each objective is defined.

Objective 1: Study of the convergent validity of PA-IADL functional tasks with cognitive assessment tests (traditional MCI detection method), hereinafter CgA, in their capacity as MCI detection tests. This contains two subordinate objectives:

Subordinate objective 1.1- To determine which PA-IADL tasks are useful to identify and differentiate participants who have been classified as MCI and participants who have been classified as Healthy by CgA tests. Because all PA-IADL tasks require cognitive strategies for their performance and cognitive skills are naturally affected by age, to achieve this subordinate objective it was considered necessary to determine two aspects. First, we assessed which tasks are useful for discriminating between MCI and Healthy individuals regardless of the age of the participants. Second, we assessed whether there were differences between MCI and Healthy participants in their performance according to age. The result of these analyses will allow us to understand which tasks have the greatest power to discriminate between MCI participants and Healthy ones, and we will call these target tasks (TT).

Subordinate objective 1.2- To evaluate the effectiveness of the TT set in classifying the participants of the sample set into clearly differentiated categories.

Objective 2: To analyze the divergences in the MCI-Healthy classification of the two methods, through the functional tasks of the PA-IADL (only the TTs) and through the CgA, in the diagnosis of MCI. This contains two subordinate objectives:

Subordinate objective 2.1- To examine the differences in the performance of the TTs of the PA-IADL between the participants that make up the 4 groups of the confusion matrix, with particular interest in the participants that the two methods classify in different groups.

Subordinate objective 2.2- To examine what characteristics are shown by the participants that the two methods classify in different groups in the variables used to carry out the matching.

The data analysis carried out to carry out each of the two subordinate objectives contained in Objective 1 and Objective 2 was as follows.

2.3.1. Data analysis in pursuit of subordinate objective 1.1

At this point, it is a priority to understand whether it is possible to study the 12 tasks together, or on the contrary, this must be done individually. The statistical advantage of multivariate tests over univariate tests is that they keep Type I error under control (Hair et al., 2006; Maxwell et al., 2017). But there is also a substantive advantage that in this particular case is very relevant, and that is that they would allow a holistic study to be carried out of the integrated PA-IADL package, in the same way that cognitive tests in the diagnosis of MCI constitute a holistic package. To do this, we began by evaluating the correlation in the performance of the 12 tasks with each other (see Table S3 in the supplementary material), and it was found that the 12 tasks are intercorrelated. Specifically, our findings were as follows: 89.39% (n=59) of the correlations were statistically significant; 45% (n=30) of the correlations had a magnitude greater than .40, the minimum value for applying a multivariate methodology (Cochran & Cox, 1957); no correlation reached the value of .80, therefore it is unlikely that there is multicollinearity (Bray & Maxwell, 1985; Tabachnick & Fidel, 1989); and Bartlett's sphericity test allows us to reject the null hypothesis that the residual covariance matrix is proportional to an identity matrix (χ2= 371.42; df=77; p=.001). For all these reasons, it is concluded that it is more appropriate to use a multivariate methodology than a univariate one (Hair et al., 2007; Maxwell & Delaney, 2004).

Thus, it was decided to carry out first a MANCOVA taking age as a covariate and then a MANOVA (EG x Age groups) introducing age as a classification variable. The two analyses provide us with complementary information. We will outline this below briefly.

Given that age has a statistically significant linear relationship with all tasks (except task 12) (see Table S3 in supplementary material), it is a priority to determine the tasks in which the difference in performance between Cases and Controls disappears when controlling for the effect of age, and the tasks in which the differences between the two groups exist even when controlling for the effect of age. This is the reason that justifies the MANCOVA, and we began by checking whether the rest of the assumptions it requires were satisfied. It was verified that there was no relationship between the covariate and the assigned independent variable (Eg), and it was also confirmed that Tasks 3 and 7 show non-parallel slopes in the groups of Cases and Controls. Based on this result, Task 12 was examined using ANOVA; for the study of Tasks 3 and 7, the regions of non-significance and significance were determined using the Johnson-Neyman technique (see Huitema, 2011) and performing the ANCOVA in the regions of significance; and the joint analysis of the remaining 10 tasks was performed using the MANCOVA following the modeling strategy proposed by Milliken & Johnson (2001). Since the difference in the execution of Tasks T2, T6, T8, and T9 between the EG disappears once the effect of age is controlled, these four tasks did not form part of the subsequent MANOVA.

Because the sample of participants covers ages between 60 and 98 years, for practical purposes, in addition to understanding whether age influences the performance of the different tasks, it is of great interest to understand whether the performance in the groups of Cases and Controls is moderated by age. To determine this, the sample was segmented into three age ranges (AR) (60-69, 70-79, and over 80 years), and a MANOVA (2x3) (EG x AR) was performed on the set of 8 tasks in which, once the effect of age was controlled, the differences between the EGs persisted. The result was examined using the Wilks Lambda statistic because it is the most robust test when the assumption of homogeneity of the covariance matrices is not satisfied (see Ates et al., 2019), as in this case [M de Box =143.53; g1=78 and 12138.7; p=.005; Levene's test detected non-homogeneous variances in tasks 4, 5, 11, and 12]. Since the interaction was statistically significant, we proceeded to study the simple effects and their corresponding mean differences (Bonferroni correction). The analysis was completed by performing a stepwise discriminant analysis to examine which combination of dependent variables had the strongest power to differentiate between the 6 groups resulting from the combination (EG x AR), paying special attention to the standardized coefficients and the magnitude and sign of the centroids. The analysis was first performed on a randomly chosen training sample (70% of cases, approx. n=50), and then on the whole sample. (It was decided not to perform the analysis on the validation sample, because 30 % of the sample, the remaining sample, is too small to be useful as a validation sample).

The reference values are the p-value (α=.05), partial eta squared (η2) (.01 small, .06 medium, and .14 large. See Cohen, 2013, Ellis, 2010), and (1-β) >.80.

To aid in the interpretation of the results obtained by both MANCOVA and MANOVA, the linear relationship between task performance and age was plotted, and also, the line of fit was calculated by iterative weighted least squares (Loess, using the Epanechnikov kernel function, where data located near a particular point receive higher weights than data that are further away).

2.3.2. Data analysis in the achievement of the subordinate objective 1.2

The previous analyses led to the conclusion that there were 8 TTs. Because the classification techniques are exploratory techniques and due to the small sample size, it was considered necessary to control the potential overfitting that could occur, and also to control the effect of the method used for classification. Thus, the analysis was carried out using 5 different methods, the hierarchical cluster procedure using two methods—the furthest neighbor method, and the Wald method— (both were chosen because they are robust to outlier scores, a probable case in such a small sample), the two-stage cluster method, the k-means method, and the discriminant method. The last two require the indicating of the number of clusters, the hierarchical procedures do not, and it is optional in the two-stage cluster analysis (in this case, it was performed without determining the number of clusters). Furthermore, the analysis was also performed by cross-validation, randomly selecting a new training sample (approximately 70% of cases, n=42). Again, due to the small size of the validation sample (n=22), the second analysis was performed on the full sample.

The percentage of coincidence was calculated between the MCI-Healthy classifications (i.e., Cases-Controls) carried out with the 5 classification techniques performed based on the TTs and the original classification carried out using the CgA tests. The magnitude of the agreement was also calculated using the Kappa coefficient.

Objective 2: To analyze the divergences in the MCI-Healthy classification of the two methods, through the functional tasks of the PA-IADL (only the TTs) and through the CgA, in the diagnosis of MCI. This contains two subordinate objectives:

Subordinate objective 2.1- To examine the differences in the performance of the TTs of the PA-IADL between the participants that make up the 4 groups of the confusion matrix, with particular interest in the participants that the two methods classify in different groups.

Subordinate objective 2.2- To examine what characteristics are shown by the participants that the two methods classify in different groups in the variables used to carry out the matching.

2.3.3. Data analysis in the achievement of subordinate objective 2.1

The decision was made to take as a reference the classification resulting from the hierarchical analysis using the Ward method and the resulting classification using the k-means method, as these were the classifications that had the highest and lowest concordance between the two methods. The differences between the 4 groups of the confusion matrix in the 8 TTs were tested using the robust Brown-Forsythe ANOVA due to the heterogeneity of the variances between the groups, in part, caused by the different size of the groups of the confusion matrix (Vallejo et al., 2010).

2.3.4. Data analysis in the achievement of subordinate objective 2.2

This was done by looking at the descriptive statistics. The analysis of the two subordinate objectives will allow us to identify the power, limitations, and vulnerability limits of the detection procedure based on cognitive tests.

The most crucial thing in a case-control study is to choose “the cases” carefully and “the controls”. If the selection is not correct, and if the matching is not accurate, there will be selection bias, and the inferences will not be valid. This is one of the reasons that justifies this section being so extensive and so detailed. Much effort was invested to correctly identify cases and controls, with the sole objective that the cases were indisputably cases and that the controls were indisputably controls. For this reason, the sample size was only 64 people in total. On the other hand, the data analysis that was carried out was in-depth and very orderly. As can be seen, each objective was carried out with different analyses as a “sensitivity analysis” to qualify the results and safeguard their replicability.

All analyses were performed using the statistical package IBM SPSS 27.

Point 5. The references are very competed, but we suggested to replace some of the oldest if possible with more recent ones from the last 3 years, to better update recent studies in the literature regarding MCI and the importance of its early detection.

Response for Point 5. The reviewer is right.

We have not eliminated any of the existing references, however, we have added 14 new references, most of them very recent. The new references added are the following:

  • Overton, M.; Pihlsgård, M.; Elmståhl, S. Diagnostic Stability of Mild Cognitive Impairment, and Predictors of Reversion to Normal Cognitive Functioning. Dement Geriatr Cogn Disord. 2019, 48 (5–6), 317–329. https://doi.org/10.1159/000506255.
  • Shimada, H.; Doi, T.; Lee, S.; Makizako, H. Reversible Predictors of Reversion from Mild Cognitive Impairment to Normal Cognition: A 4-Year Longitudinal Study. Alz Res Therapy. 2019, 11 (1), 24. https://doi.org/10.1186/s13195-019-0480-5.
  • Katabathula, S.; Davis, P. B.; Xu, R. Comorbidity‐driven Multi‐modal Subtype Analysis in Mild Cognitive Impairment of Alzheimer’s Disease. Alzheimer’s & Dementia. 2023, 19 (4), 1428–1439. https://doi.org/10.1002/alz.12792.
  • Liu, H.-Y.; Tsai, W.-C.; Chiu, M.-J.; Tang, L.-Y.; Lee, H.-J.; Shyu, Y.-I. L. Mild Cognitive Impairment in Combination with Comorbid Diabetes Mellitus and Hypertension Is Negatively Associated with Health-Related Quality of Life among Older Persons in Taiwan. Qual Life Res. 2019, 28 (5), 1281–1291. https://doi.org/10.1007/s11136-019-02101-3.
  • Paul, A.; Padmanabhan, D.; Suresh, V.; Nayagam, S.; Kartha, N.; Paul, G.; Vijayakumar, P. Analysis of Neuropathological Comorbid Conditions in Elderly Patients with Mild Cognitive Impairment in a Tertiary Care Center in South India. J Family Med Prim Care. 2022, 11 (4), 1268. https://doi.org/10.4103/jfmpc.jfmpc_1094_21.
  • Miyamatsu, N.; Kiyohara, M.; Kawahara, M.; Morino, K.; Miyazawa, I.; Fujita, Y.; Shima, A.; Maegawa, H.; Ogita, M. Interaction of comorbid hypertension and sex on occurrence of mild cognitive impairment in outpatients with diabetes mellitus: a 6-year follow-up study. Journal of Hypertension 2023, 41 (Suppl 3), e161. https://doi.org/10.1097/01.hjh.0000940488.58685.fd.
  • Beech, B. F.; Sumida, C. A.; Schmitter-Edgecombe, M. Real-World Compensatory Strategy Use in Community-Dwelling Mid-Life and Older Adults: An Evaluation of Quality. The Clinical Neuropsychologist 2023, 1–24. https://doi.org/10.1080/13854046.2023.2209927.
  • Ross, S. D.; Rodriguez, F. S. Usability of a Memory Aid Handbook for Older People with Subjective Cognitive Impairment—An Explorative Pilot Study. Int J Geriat Psychiatry 2023, 38 (8), e5989. https://doi.org/10.1002/gps.5989.
  • Terao, C. M.; Pishdadian, S.; Moscovitch, M.; Rosenbaum, R. S. Ask How They Did It: Untangling the Relationships Between Task-Specific Strategy Use, Everyday Strategy Use, and Associative Memory; preprint; PsyArXiv https://doi.org/10.31234/osf.io/y6awu.
  • Weakley, A.; Weakley, A. T.; Schmitter-Edgecombe, M. Compensatory Strategy Use Improves Real-World Functional Performance in Community Dwelling Older Adults. Neuropsychology 2019, 33 (8), 1121–1135. https://doi.org/10.1037/neu0000591.
  • Joly-Burra, E.; Van der Linden, M.; Ghisletta, P. A Mixed-Method Study on Strategies in Everyday Personal Goals among Community-Dwelling Older Adults. Gerontology. 2020, 66 (5), 484–493. https://doi.org/10.1159/000508824.
  • Makino, K.; Lee, S.; Bae, S.; Shinkai, Y.; Chiba, I.; Shimada, H. Relationship between Instrumental Activities of Daily Living Performance and Incidence of Mild Cognitive Impairment among Older Adults: A 48-Month Follow-up Study. Archives of Gerontology and Geriatrics. 2020, 88, 104034. https://doi.org/10.1016/j.archger.2020.104034.
  • Cloutier, S.; Chertkow, H.; Kergoat, M.; Gélinas, I.; Gauthier, S.; Belleville, S. Trajectories of Decline on Instrumental Activities of Daily Living Prior to Dementia in Persons with Mild Cognitive Impairment. Int J Geriat Psychiatry. 2021, 36 (2), 314–323. https://doi.org/10.1002/gps.5426.
  • Divers, R. M.; De Vito, A. N.; Pugh, E. A.; Robinson, A.; Weitzner, D. S.; Calamia, M. R. Longitudinal Predictors of Informant-Rated Everyday Function in Mild Cognitive Impairment. J Geriatr Psychiatry Neurol 2023, 36 (1), 18–25. https://doi.org/10.1177/08919887221093360.

Reviewer 3 Report

Comments and Suggestions for Authors

Dear Authors.

The manuscript is presented forcefully both in the foundations that support the need for the study, the methodology that collects and processes the data, and the presentation of the results, in a relevant topic to evaluate mild cognitive impairment in older adults with a test functional appropriate to your level of activities. Monitoring of this line is necessary. Congratulations on that.

However, some minor observations, especially in the discussion and conclusions, are presented as suggestions that could contribute to the presentation of this publication in the academic concert:

In the introduction there is a pertinent approach to the topic that is supported by recent literature and also exposes the variables to be considered, substantiating the problem to be addressed clearly by exposing the two general objectives of the research. No sugest.

The methodology, suggests adding the approval of an Ethics Committee for this procedure, although presents the sample selection considering reliable inclusion/exclusion criteria and the informed consent procedure. The selection of tests that include cognitive and functional evaluations, especially those relevant to episodic memory, which is key to determining mild cognitive impairment, are relevant and maintain reliable and valid statistical data. They were also applied in 4 different instances, which avoids fatigue bias in older adults. This is notable.

The selection of the statistical analysis is pertinent taking into account the complexity of evaluating the relationship between the variables studied. They managed to present an analysis consistent with the objectives proposed in the introduction to the manuscript. Segmentation by age groups among older adults was also relevant.

Line 23-24:Avoid adjectives like these in the text.

Line 57: The stated statement requires bibliographical citation.

In the discussion and conclusions some more in-depth suggestions:

Line 606-631. It is advisable to support the aspects pointed out by other authors, in addition to Tomaszewski, such as DOI 10.1007/s00415-015-7662-2 or DOI 10.1017/S1355617719000912, among others, because the discussion almost only revolves around their results and procedures.

The same thing as previously mentioned occurs in point 4.2 of the discussion. A discussion in light of what has been published on the subject enriches the analysis of the results of this manuscript. For example, see, among others DOI: 10.1007/s11065-015-9312-y

On the other hand, what is linked to the activities of daily living and its relationship with mild cognitive impairment, especially with what is expressed in terms of age, requires further discussion in light of other findings, in addition to Libon or Jekel referred to, for example, but not limited to: DOI 10.1016/j.archger.2020.104034; DOI 10.1080/13803395.2011.614598;

The key variables of this study that affect the results are important to discuss in light of other research. Furthermore, since the sample is small, it suggests contrasting it with other findings. Thus the conclusion of this investigation will be pertinently supported. 

Comparisons with other cognitive tests should also be cited to establish the contrast with the contribution of this research, preparing the final synthesis and the contribution of this paper to the scientific world.

Line 727 to 742: Suggests that the wording of the conclusions should refer to the findings obtained and previously discussed, especially the benefits of the PA-AIVD, and not return to the discussion already carried out on other cognitive tests that evaluate mild cognitive impairment. This perspective with an approach that quantifies the qualitative functioning is what is valuable in my view as a valuable contribution to this research.

Author Response

Response to Reviewer 3 Comments

We want to thank the reviewer for the careful and critical reading of this article because their considerations have contributed to substantially improving the research work presented in this manuscript, and the presentation of the manuscript itself.

In response to the reviewer's requests, a large part of the article has been modified; without a doubt, it is now much clearer, more informative, and easier to read, and the motivation and development of the proposed procedure are much better understood.

We respond in detail to each of your wise suggestions as follows.

-We highlight in black as Point 1, Point 2, etc., the suggestions made by the reviewer.

-We highlight the answers to each of the previous points in red. As follows: Response for Point 1, Response for Point 2, etc.

-In the new version of the scientific article, all the changes made have been introduced and explained. The new text that has been added has been highlighted in blue so that the changes made can be seen more clearly.

-In this response document to the reviewer, some parts of the paper are shown literally. These portions of text that are displayed verbatim are displayed tabulated. The reviewer can verify that they appear identically in the new version of the paper.

Comments and Suggestions for Authors

The manuscript is presented forcefully both in the foundations that support the need for the study, the methodology that collects and processes the data, and the presentation of the results, in a relevant topic to evaluate mild cognitive impairment in older adults with a test functional appropriate to your level of activities. Monitoring of this line is necessary. Congratulations on that.

However, some minor observations, especially in the discussion and conclusions, are presented as suggestions that could contribute to the presentation of this publication in the academic concert:

Point 1. In the introduction there is a pertinent approach to the topic that is supported by recent literature and also exposes the variables to be considered, substantiating the problem to be addressed clearly by exposing the two general objectives of the research. No sugest.

Point 2. The methodology, suggests adding the approval of an Ethics Committee for this procedure, although presents the sample selection considering reliable inclusion/exclusion criteria and the informed consent procedure. The selection of tests that include cognitive and functional evaluations, especially those relevant to episodic memory, which is key to determining mild cognitive impairment, are relevant and maintain reliable and valid statistical data. They were also applied in 4 different instances, which avoids fatigue bias in older adults. This is notable.

Point 3. The selection of the statistical analysis is pertinent taking into account the complexity of evaluating the relationship between the variables studied. They managed to present an analysis consistent with the objectives proposed in the introduction to the manuscript. Segmentation by age groups among older adults was also relevant.

Response for Point 1 to Point 3.

Much effort has been invested in this research to care for and guarantee methodological quality in all phases of the research process. Therefore, we want to thank the reviewer for this article's careful and critical reading and the positive feedback.

Point 4. Line 23-24:Avoid adjectives like these in the text.

Response for Point 4: The reviewer is right. The following phrase has been removed at the request of reviewer 3: "have a remarkable relevance and"

Point 5. 2-Line 57: The stated statement requires bibliographical citation.

Response for Point 5: The reviewer is right. The following quotes have been introduced at the request of reviewer 3:

  • Overton, M.; Pihlsgård, M.; Elmståhl, S. Diagnostic Stability of Mild Cognitive Impairment, and Predictors of Reversion to Normal Cognitive Functioning. Dement Geriatr Cogn Disord. 2019, 48 (5–6), 317–329. https://doi.org/10.1159/000506255.
  • Shimada, H.; Doi, T.; Lee, S.; Makizako, H. Reversible Predictors of Reversion from Mild Cognitive Impairment to Normal Cognition: A 4-Year Longitudinal Study. Alz Res Therapy. 2019, 11 (1), 24. https://doi.org/10.1186/s13195-019-0480-5.

Point 6. 3-Line 606-631. It is advisable to support the aspects pointed out by other authors, in addition to Tomaszewski, such as DOI 10.1007/s00415-015-7662-2 or DOI 10.1017/S1355617719000912, among others, because the discussion almost only revolves around their results and procedures.

Response for Point 6: The reviewer is right. The following paragraph has been introduced in section 4.1 at the request of reviewer 3:

The use of compensatory strategies in everyday life has been the subject of studied [75, 76, 77] and results suggest that their use seems to improve performance in everyday tasks in older people with poorer cognitive performance [78]. Joly-Burra et al. [79]. analyzed the selective optimization by compensation (SOC) model in a sample of people over 65 years of age and concluded that the use of compensatory strategies was more directly related to the cognitive performance of the participants and not so much to age. It would be of interest in the future to delve into how the use of compensatory strategies affects daily tasks in people with MCI.

See also the joint response to points 9, 10 and 11.

Point 7. The same thing as previously mentioned occurs in point 4.2 of the discussion. A discussion in light of what has been published on the subject enriches the analysis of the results of this manuscript. For example, see, among others DOI: 10.1007/s11065-015-9312-y

Response for Point 7: The reviewer is right. The following paragraph has been introduced in section 4.2 at the request of reviewer 3:

To our knowledge, how healthy and MCI individuals behave about functional performance in IADL according to age range has not been studied. Longitudinal studies have shown a progressive increase with age of limitations in complex daily functions in people with MCI [80, 81] and people with prodromal signs of MCI [5]. These changes in everyday function are associated with cognitive processes such as processing speed, executive function, and memory, especially in financial tasks [82]. In this study, discriminant analysis allows us to observe the functional performance of the PA-IADL of healthy people and people with MCI according to age range.

See also the joint response to points 9, 10 and 11.

Point 8. On the other hand, what is linked to the activities of daily living and its relationship with mild cognitive impairment, especially with what is expressed in terms of age, requires further discussion in light of other findings, in addition to Libon or Jekel referred to, for example, but not limited to: DOI 10.1016/j.archger.2020.104034; DOI 10.1080/13803395.2011.614598;

Response for Point 8: The reviewer is right. Indeed, age is an imponderable variable; in fact, we have studied it in this research as a covariate and as a moderating variable.

So that this aspect so well brought up by the reviewer is made evident, we have added in section 4.2 the following paragraph:

Having found that age is an imponderable variable that affects performance in tasks. Having found that age is a moderating variable and, therefore, the differences in task performance between Cases and Controls are greater or lesser in the three arbitrarily formed ARs. Having found that some tasks are more affected than others in the different ARs and that the difference between Cases and Controls is not of the same magnitude in all of them, nor does it appear in the same ARs, it is worth asking what exists beyond, what exists regardless of the age that explains these differences, what cognitive abilities are involved in the tasks that explain these results? In the next point, we try to discuss this aspect.

See also the joint response to points 9, 10 and 11.

Point 9. The key variables of this study that affect the results are important to discuss in light of other research. Furthermore, since the sample is small, it suggests contrasting it with other findings. Thus the conclusion of this investigation will be pertinently supported. 

Point 10. Comparisons with other cognitive tests should also be cited to establish the contrast with the contribution of this research, preparing the final synthesis and the contribution of this paper to the scientific world.

Point 11. Line 727 to 742: Suggests that the wording of the conclusions should refer to the findings obtained and previously discussed, especially the benefits of the PA-AIVD, and not return to the discussion already carried out on other cognitive tests that evaluate mild cognitive impairment. This perspective with an approach that quantifies the qualitative functioning is what is valuable in my view as a valuable contribution to this research.

Response for Point 9 to Point 10. The reviewer is right. Considering all the excellent suggestions of Reviewer 3, we decided to redraft the section 4 Discussion. In the new manuscript, we call it 4. Discussion and Conclusions. This section answers to the suggestions contained in points 9, 10, and 11 but includes all the contextualization of the answer to the suggestions contained in points 6, 7, and 8 too.

  1. Discussion and conclusions

A Case-Control study has been carried out with two fundamental objectives: firstly, to analyze to what extent the performance in the functional tasks of the IADL-PA and the cognitive tests converge in their ability to detect MCI in people without pathologies, without moderate or severe cognitive impairment, and who live independently in the community and, secondly, analyze the divergences in the classification of the two detection methods and identify their limits of vulnerability, as well as determine the power and limitations of both procedures.

Both objectives were aimed at evaluating the convergent validity of performance-based activities of daily living assessed by PA-IADL test in relation to traditional (standard) cognitive assessment to identify older adults with mild cognitive impairment, and which gives a name to the title of the article. In this section, we are going to discuss the results found in the order in which the data analysis was planned, and in this way, in addition to discussing the results found, we discover the new hypotheses that are proposed in light of these and also the conclusions generated. Thus, we organize the most significant results and their corresponding analysis into 4 points.

4.1. The influence of age on task performance (IADL-PA) and the differences between Cases and Controls after controlling for the effect of age.

           Age has been found to influence the performance of all tasks in the Cases and Controls. However, the influence of age is not the same in all of them. Its effect could be summarized in three points.

One, Cases and Controls differ statistically significantly in task 12. However, age does not explain this difference at all.

Two, age is the only variable that explains the difference between Cases and Controls in the execution of tasks T2, T6, T8, and T9. For this reason, once the effect of age is controlled, the differences between Cases and Controls in these four tasks disappear.

Third, age is a variable that explains in some that the difference between Cases and Controls in executing tasks T1, T3, T4, T5, T7, T10, and T11. In fact, once the effect of age is controlled, the differences between Cases and Controls persist. Therefore, other variables that make the Cases and Controls different explain these differences.

Three conclusions emerge from these results.

First, age is not a decisive or significantly relevant predictor in classifying healthy individuals and those with mild cognitive impairment. Abundant studies support the notion that MCI can manifest as a comorbidity associated with other pathologies, diabetes, cancer, etc., and not necessarily as a product of age [70-73].

Second, although age explains the differences between the Cases and Controls in tasks T2, T6, T8, and T9, it must not be ruled out that the small sample size is responsible for the lack of test power to detect differences between both groups once age was controlled. We think this because, if we look at Figure S1 (in supplementary material), we see that, in these four tasks, both groups have apparently different behavior in the age range between approximately 70-79 years and a behavior similar to older age and at a younger age (except T2 perhaps). A possible explanation for these results could be that tasks T6 and T9 are tasks in which administrative and banking documentation must be handled and have a crystallized component, that is, they require domain-specific knowledge acquired through practice throughout life. They are, therefore, tasks in which people with mild cognitive impairment can make up for their deficits with the acquired knowledge resulting from their daily practice through compensatory strategies [74].

           The use of compensatory strategies in everyday life has been the subject of studied [75, 76, 77] and results suggest that their use seems to improve performance in everyday tasks in older people with poorer cognitive performance [78]. Joly-Burra et al. [79]. analyzed the selective optimization by compensation (SOC) model in a sample of people over 65 years of age and concluded that the use of compensatory strategies was more directly related to the cognitive performance of the participants and not so much to age. It would be of interest in the future to delve into how the use of compensatory strategies affects daily tasks in people with MCI.

Third, if age does not explain the differences in execution of the Cases and Controls in task 12, and if age is not the only variable that explains the differences in execution between the Cases and Controls in tasks T1, T3, T4, T5, T7, T10, and T11, the following hypotheses arise: is age a moderating variable? That is, are the differences in performance between Cases and Controls in these tasks moderated by age? If we segment age in 10-year intervals (for example), would the differences between Cases and Controls be of the same magnitude in the same age ranges in all tasks? Is there a critical age range in the debut of MCI?

4.2. The differential behavior between Healthy and MCI individuals depending on the age range

           The results found with the MANOVA allow us to conclude that performance in tasks T1, T3, T4, T5, T7, T10, T11, and T12 is different in Cases and Controls in the three arbitrarily formed age ranges. From these results, five conclusions emerge that must be discussed, and new hypotheses are proposed that open new lines of research. The following.

First, age has a moderating effect on the performance of tasks T1, T3, T4, T5, T7, T10, T11, and T12, and therefore, performance in them is performed with different solvency in healthy and in people with MCI in the three ARs. As highlighted in the introduction, this aspect has been widely documented.

Second, at 60-69 years (AR1), people with MCI show a performance similar to that of healthy people aged 70-79 years (AR2). That is, people with MCI appear to experience a 10-year delay in AR1 compared to healthy people. Furthermore, it has been proven that the differences between healthy people and MCI people are already in AR1 (η2=.28) and that task T3 is the only task that has had the strength to detect these differences (possibly task 7 could also have detected this difference. However, it is possible that the small sample size was responsible for the lack of test power). However, in this research, it has not been possible to know at what age these differences begin to be significant. We could call this critical age the debut age in DCL. This aspect could be known through a cohort design.

Third, at the age of 70-79 years (AR2) individuals with MCI experience a marked decline in their functional capacity. Performance on tasks T1, T3, T4, T7, T10, T11 and T12 is much more impaired than the performance of healthy people over 80 years of age (AR3). That is, people with MCI appear to experience a delay greater than 10 years in AR2 compared to healthy people. It is in this age group where the difference in performance between healthy people and MCI people is most remarkable (η2=.43). Arguably, this age is a critical age, not in the debut of MCI, but in the flowering of MCI. From that age onwards, the decline in MCI people continues, but slowly.

Fourth, there is also a critical age for deterioration in task performance in healthy people. This critical age is undeniably observed in AR3. Just as in people with MCI, once significant decline occurs, decline continues, but it is possible, more slowly. However, in no case is this outbreak as strong as that experienced by people with MCI. In AR3 there are differences between Cases and Controls in tasks T1, T4, T5, T10, T11 and T12 (η2=.28), but the magnitude of the differences between Cases and Controls is much smaller than that observed in AR2.

Fifth, the discriminant analysis reveals the same result as the MANOVA (replicates the result) and offers complementary information. It is enough to observe the value of the Cases and Controls centroids to appreciate, with the critical prudence that the good work of a scientist requires, that the hypotheses proposed as a result of the results discussed in the second, third, and fourth points above are plausible. In the set of 64 participants, the distance between the centroids of Cases and Controls in AR1, AR2, and in AR3 is .554, 1.8, and 1.28, respectively.

           To our knowledge, how healthy and MCI individuals behave about functional performance in IADL according to age range has not been studied. Longitudinal studies have shown a progressive increase with age of limitations in complex daily functions in people with MCI [80, 81] and people with prodromal signs of MCI [5]. These changes in everyday function are associated with cognitive processes such as processing speed, executive function, and memory, especially in financial tasks [82]. In this study, discriminant analysis allows us to observe the functional performance of the PA-IADL of healthy people and people with MCI according to age range.

           Having found that age is an imponderable variable that affects performance in tasks. Having found that age is a moderating variable and, therefore, the differences in task performance between Cases and Controls are greater or lesser in the three arbitrarily formed ARs. Having found that some tasks are more affected than others in the different ARs and that the difference between Cases and Controls is not of the same magnitude in all of them, nor does it appear in the same ARs, it is worth asking what exists beyond, what exists regardless of the age that explains these differences, what cognitive abilities are involved in the tasks that explain these results? In the next point, we try to discuss this aspect.

4.3. Differences between Cases and Controls by age group and type of task. The cognitive demands required by the tasks could explain the differences in performance between Cases and Controls.

It is plausible to hypothesize that the PA-IADL tasks that would best discriminate between Controls and Cases would be those that combine working memory, executive processes, and episodic memory [27, 38, 42, 44].. Let's consider the three arbitrarily formed age ranges (60-69, 70-79, 80 and over) and the type of task. We can see that in tasks T1, T4, T10, T11, and T12, the Cases have a worse performance than the Controls from the age of 70. More cognitively demanding tasks are affected to a greater extent, such as task T1 (filling pill boxes) or task T10 (meal preparation), in which several cognitive processes are involved (cognitive flexibility, working memory, planning, and reasoning). Still, those in which only episodic memory is involved, such as task T12 (shopping for food), are affected to a lesser extent. Several studies have pointed out that people with MCI who are affected several cognitive domains perform worse in everyday tasks than people with MCI that affected in a single domain [7, 83]. On the contrary, in tasks T3 (medication control) and T7 (prospective memory without cues) the deterioration pattern by age is different; it is very pronounced at AR1, but at advanced ages (80 and older) we found no differences between MCI and Healthy individuals; That is, at younger ages (60-70 years) MCI individuals have a much worse performance than Healthy people or, in other words, the deterioration begins at early ages. These tasks are very cognitively demanding. In the case of task T7 the person must remember to deliver a document after 15 minutes without any help and in task T3, the person must calculate how many days there are pills left in the rheumatism bottle and to do so, they must perform up to six different well-planned steps (executive functioning) and different calculations (working memory). In this sense, we should not forget that when we perform an IADL, we set in motion different tasks simultaneously (in our case, steps) with several cognitive processes working together [6, 84]. Thus, tasks T3 and T7 allow the detection of people with MCI at an earlier age. This is very important for clinical practice since the earlier an MCI situation is detected, the earlier intervention can take place, and this will positively affect the evolution of cognitive functioning and the ability to perform daily activities autonomously. However, it is necessary to design new tasks of a similar nature to analyze convergent and predictive validity. Finally, in tasks T1 (filling pill boxes), T5 (bank direct debit), and T11 (bus route), the significant gap between MCI and Healthy individuals occurs at advanced ages (80 years and older). These tasks involve cognitive processes such as working memory, planning, and cognitive flexibility, for which training could be given to delay the deterioration in functional capacity as much as possible.

           Once the eight tasks responsible for the most pronounced differences between Cases and Controls have been identified, and the cognitive resources involved in them have been identified, could the score achieved in the eight tasks serve to identify who are Cases and who are Controls?

4.4. Convergence in the classification of Cases and Controls of PA-IADL test and traditional (standard) cognitive assessment.

The five exploratory classification methods executed, with the predictor variables being performance in the eight most discriminative tasks, return the same result (strong replication). The 5 achieved a very high convergence in the classification of the participants with the traditional (standard) cognitive assessment, between 76.56% (K-means) and 81.24% (Hierarchical Cluster procedure using the Ward method). The difference in means between the healthy people and people with MCI whom the two procedures have identified is very relevant in each of the eight tasks.

Therefore, two things can be concluded. One that the convergent validity of performance-based activities of daily living assessed by PA-IADL test about traditional (standard) cognitive assessment to identify older adults with mild cognitive impairment has been demonstrated, and two, given that each of the eight tasks is useful to capture the differences between Cases and Controls, given that in each task there are several different cognitive abilities involved, given that these cognitive abilities can be identified in the various “steps” into which each task is segmented, the identification of Cases and Controls using PA-IADL could be extremely useful to know which cognitive ability is most affected in a person, which cognitive ability deteriorates first, which cognitive abilities are most resilient, and much more of hypotheses that can be tested through appropriately planned experimental research to test the desired effect. It is also extremely useful because the alarm that “something is happening” in a person can be identified more quickly, even by the person's close relatives, and early intervention can be carried out to stop or slow down the process.

Another very striking result has to do with the performance in the tasks of the participants in which there was no consensus in the classification. In detail:

On the one hand, the six individuals classified as MCI by the k-means cluster method based on the PA-IADL, and classified as Healthy by the cognitive assessment method have statistically equal performance in all tasks to those classified as MCI by both methods. These six people were over 80 years of age. Although they were cognitively well according to the cognitive tests, however at that age, different processes are affected that have a decisive influence on daily functioning that the cognitive tests cannot detect. One might think that cognitive status is not a good predictor of daily functioning at that age or that the PA-IADL tasks are more accurate in discrimination because they involve the combination of different cognitive processes that are evaluated together in action and that at that age are more committed [85].

On the other hand, the nine people classified as Healthy by the k-means cluster method based on the PA-IADL and as MCI by the cognitive assessment have a performance in all tasks statistically equal to those classified as Healthy by both methods. These nine people were younger, and although they performed worse in the cognitive tests than their normative group, this deterioration did not seem to affect the resolution of everyday tasks. That is to say, they presented fewer difficulties in solving daily tasks in which it is necessary to set in motion different cognitive processes simultaneously and which, at these younger ages, are not so affected due, perhaps, to less brain slowing.

From our point of view, these results invite us to hypothesize that PA-IADL [85] is more accurate than the traditional (standard) cognitive assessment to identify people with MCI and Healthy people.

Traditionally, cognitive function is assessed through cognitive tests. One of these tests is the MMSE, which has problems in detecting mild cognitive impairment, specifically, a ceiling effect and lower sensitivity [86, 87].  Our study used the MMSE to rule out people with moderate or severe cognitive impairment. We applied a broad battery of cognitive tests to analyze the relationship between cognitive processes and daily functioning, and this battery served, in turn, to detect people with mild cognitive impairment. Another test to evaluate cognitive function in older people is the MoCA. Some studies highlight the effectiveness of MoCA in detecting MCI [88, 89]. However, in some MoCA tasks, older people must use decontextualized material and apply formal logic, where their personal experiences and stored knowledge are useless. In this type of cognitive assessment, it is easy to demonstrate the losses associated with normal aging without pathology; hence, people without cognitive impairment could be classified as MCI. In contrast, assessing everyday functioning through performance-based tests provides insight into how older people cope with contextualized tasks, with a logic based on the every day and what these people do in their daily lives.

Since cognitive processes are involved in everyday activities, the assessment of day-to-day functioning also makes it possible to analyze cognitive functioning and detect problems in this functioning from a more realistic perspective, avoiding issues in the classification of people with MCI and Healthy individuals. Thus, and in light of these results, it could be concluded that this is true.

4.5. Limitations

This study has two important limitations. One, the influence that the training or educational level has on the execution of the tasks has not been evaluated. Different studies indicate that the training or academic level is a variable strongly related to cognitive functioning [90, 91], and therefore, as has been evidenced in this research, also will be related to performance in the execution of tasks. However, this variable was used to match Cases and Controls, and therefore, the results found are free of systematic bias due to the training or educational level of the participants. Two, the small sample size. Although the data analysis carried out has allowed us to conclude that each task behaves differently in the interaction (Age Range x EG Case/Control), and all the simple effects have been statistically significant with a high effect size, we think that, perhaps, the small number of subjects does not allow us to find statistically significant differences in task 7 (prospective memory without cues) in AR1, so in future research the sample size should be planned a priori [92], at least to test the most difficult hypotheses to test.

4.6. Summary of most relevant contributions and prospective

However, despite these limitations, this study highlights the valuable contribution of IADL-PA as a test to assess functional capacity in the detection of individuals with MCI. On the one hand, performance in the different tasks allows us to predict, regardless of age, whether a person is healthy or has MCI, which represents an added value of the PA-IADL to the extent that it will allow us to differentiate a deterioration in the processes cognitive functions involved in daily tasks of a decline in cognitive functioning associated with age, especially in older adults. On the other hand, the set of functional tasks is capable of detecting MCI with greater precision because each of the tasks involves several cognitive competencies that in combination determine whether a task is performed successfully or not. Cognitive psychometric tests can never capture this aspect. Each of these cognitive tests participates in detection, as does all of them, but it is impossible to quantify what the interaction of several of them would mean, as in the case of functional tasks [93]. 

In addition to these two aspects that we consider to be of utmost relevance, in the writing of this section dedicated to the discussion and conclusions, some new hypotheses have been highlighted that arise in light of these results and that raise new lines of research, some of which should be resolved through observational cohort research and others through experimental research.
